# Partitioning for Intrinsic Model Inversion Resistance in Collaborative Inference

**Rongke Liu**[1]  **Youwen Zhu**[1]  **Lei Zhou**[1]  **Xianglong Zhang**[2]  **Dong Wang**[3]

## Abstract

In collaborative inference (CI), transmitting intermediate representations $Z$ from edge devices enables model inversion attacks (MIA) that reconstruct the original inputs $X$, while existing defences mainly perturb shallow-layer $Z$ at the cost of utility. We instead ask: *where should an edge–cloud model be partitioned to obtain intrinsic resistance to MIA?* We challenge the intuition that depth is the driver of MIA resistance, and show that depth is sufficient only insofar as it enables a representational transition; this transition is necessary for *intrinsic* resistance and is marked by an abrupt rise in the lower bound of $H(X \mid Z)$. Correspondingly, the decisive variance term in the entropy bound shifts from a global variance to the intra-class mean-squared radius $R_c^2$ rather than dimensionality alone, yielding an $R_c^2$-based criterion to locate the transition zone, or identify it post hoc from MIA outcomes, which we term the *Golden Partition Zone* (GPZ). We further explain how $R_c^2$ evolves during training and show that it can be controlled through the label distribution; we refer to this controllable dynamic behavior as the *Neural Vortex*, an analysis-backed explanatory concept. Across four representative deep vision models, partitioning at the GPZ yields over 4× higher reconstruction MSE compared to shallow splits; under entropy and inversion-model enhancements, decision-level representations provide 66% stronger resistance than feature-level ones, and we further observe that data type affects both the transition boundary and reconstruction.

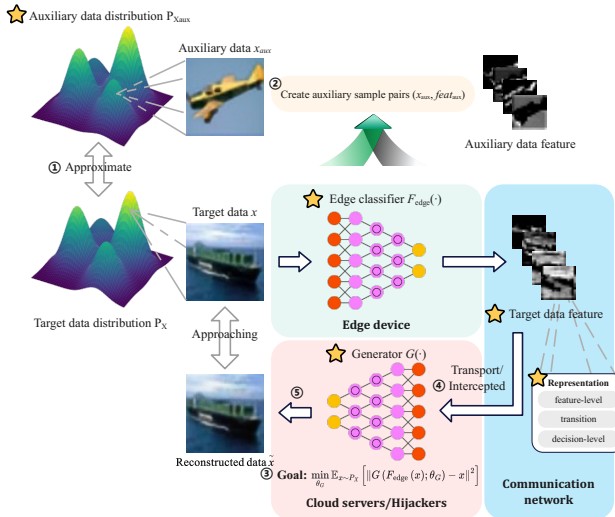

*Figure 1.* Paradigm of MIA in collaborative inference, with ⋆ indicating the parts covered in our contributions.

## 1. Introduction

Collaborative inference (CI) (Kang et al., 2017) has become a widely adopted paradigm in edge–cloud computing, and is increasingly the backbone of artificial intelligence (AI) services, where a deep neural network (NN) is partitioned between an edge device and a cloud server. This design enables efficient inference and raw data protection by allowing the edge to perform early computations while the cloud processes the remaining. The method supports real-time performance and has seen broad adoption in UAVs (Qu et al., 2024), IoT systems (Shlezinger & Bajić, 2022), and private cloud computing (PCC) (Engineering & Architecture, 2024). With the continued rise of 5G, IoT, and AI, CI is expected to become even more prevalent. However, recent studies have shown that the transmitted information can be exploited by adversaries to reconstruct the original input through model inversion attacks (MIA) (He et al., 2019).

As illustrated in Fig. 1, MIA under CI aims to train a generator using auxiliary data that approximates the target distribution, such that the generator can approximate the inverse mapping of the edge-side model (He et al., 2019). Unlike centralized MIA (Zhang et al., 2020; Liu et al., 2024b), which typically recover class-level distributions, collaborative inference's MIA (He et al., 2019; Chen et al., 2024)

---

[1]School of Computer Science and Technology, Nanjing University of Aeronautics and Astronautics, Nanjing, China [2]Institute for Network Sciences and Cyberspace, Tsinghua University, Beijing, China [3]School of Cybersecurity, Hangzhou Dianzi University, Hangzhou, China. Correspondence to: Youwen Zhu <zhuyw@nuaa.edu.cn>.

*Proceedings of the 43$^{rd}$ International Conference on Machine Learning*, Seoul, South Korea. PMLR 306, 2026. Copyright 2026 by the author(s).

focuses on reconstructing individual samples, posing a more fine-grained privacy risk for AI services.

Most existing MIA-in-CI studies focus on shallow layers, where transmitted information preserves rich semantics and remains highly informative about the inputs, enabling reconstruction. Existing defences largely fall into two lines: (i) information perturbation (e.g., noise (Wang et al., 2022), masks (Ding et al., 2024), deeper models and information-theoretic bottlenecks (Xia et al., 2025; Duan et al., 2023)), which can mitigate reconstruction but often comes at the cost of utility and generalization; and (ii) cryptographic approaches (e.g., homomorphic encryption (Zhu et al., 2025)), which secures communication and computation but leaves intrinsic leakage unchanged and incurs overhead for edge. Accordingly, our goal is not to propose another incremental defence and compete on the privacy–utility trade-off; instead, we raise a more upstream question: *where should an edge–cloud model be partitioned to obtain intrinsic resistance to MIA?* Identifying such a zone could enable more effective and diverse defence strategies in the future.

In this work, we challenge the intuition that depth drives MIA resistance (He et al., 2019; Ding et al., 2024). Experiments on Vision Transformer (ViT) (Dosovitskiy et al., 2021) reveal that depth alone fails to ensure protection; inversion is suppressed when the mutual information $I(X, Z)$ between the input $X$ and intermediate information $Z$ drops sufficiently. On residual architectures (He et al., 2016) such as IR-152 and ResNet-50, we observe that within a certain depth range, deeper layers can even mildly improve inversion quality. From an information-theoretic perspective, we derive the lower bound on the conditional entropy $H(X \mid Z)$ and show that it increases abruptly when the representation transitions from feature-level to decision-level; correspondingly, the decisive variance term shifts from global total variance to the intra-class mean-squared radius $R_c^2$ rather than dimensionality alone. This implies that *depth is sufficient only insofar as it enables a representational transition, whereas the transition itself is necessary for intrinsic resistance.* Motivated by this, we use $R_c^2$ to locate (or infer post hoc from MIA outcomes) the transition zone, termed the *Golden Partition Zone* (GPZ). We further explain how $R_c^2$ evolves during training and show that it can be controlled through the label distribution; we refer to this controllable dynamic behavior, formulated as an analysis-backed explanatory concept, as the *Neural Vortex*.

Based on these insights, we conduct experiments on four representative deep vision models, IR-152, ResNet-50 (He et al., 2016), VGG19 (Simonyan & Zisserman, 2014), and ViT (Dosovitskiy et al., 2021), and demonstrate that partitioning at the GPZ yields over 4× higher reconstruction MSE compared to shallow splits. We further introduce targeted enhancement strategies, including mathematically

grounded and NN–based techniques to enrich $Z$, as well as a progressive attention mechanism that better approximates the behavior of $H(X \mid Z)$. Experiments demonstrate that decision-level representations provide, on average, 66% higher resistance compared to feature-level ones. Finally, we study data dependence and find that the transition boundary shifts across data types rather than occurring at a fixed network depth. Moreover, higher intra-class diversity typically yields stronger post-transition resistance to inversion.

Our main contributions are summarized as follows:

- We recognize and actively locate the GPZ for CI partitioning, where the representational transition triggers an abrupt increase in the lower bound of $H(X \mid Z)$, challenging the depth-based safety intuition. (Sec. 3.1)

- We characterize how $R_c^2$ evolves toward and beyond the GPZ during training, how label-distribution design shapes this evolution, and how it is associated with inversion resistance; we refer to this analysis-backed explanatory concept as the *Neural Vortex*. (Sec. 3.2)

- We validate the intrinsic MIA resistance of GPZ partitioning on four representative vision models, and further stress-test it with higher-entropy $Z$ and stronger inversion models. (Sec. 4.2)

- We show that data shift the transition boundary and affect reconstruction quality, shaping both GPZ localization and inversion outcomes. (Sec. 4.2)

## 2. Background

We follow the standard threat model for MIA-in-CI, focusing on images and deep vision models.

### 2.1. Collaborative Inference Setting

In collaborative inference, the information flow follows $x \xrightarrow{f_{\text{edge}}} z \xrightarrow{f_{\text{cloud}}} y$, where the deep model is split into an edge-side component $f_{\text{edge}}$ and a cloud-side component $f_{\text{cloud}}$. The edge device, controlled by the end user, processes the sensitive input $x$. This edge component performs early-layer computations and transmits intermediate information $z$ to the cloud. The cloud server, managed by a service provider, completes the remaining computation and returns the prediction $y$, either for direct feedback or downstream processing (Liu et al., 2024a).

In this setup, the cloud may act as the adversary, and an attacker may also eavesdrop on the transmission of $z$; in all cases, the adversary does not access the raw input $x$. Importantly, transport-layer encryption such as TLS (Dierks & Rescorla, 2008) only protects data in transit from plaintext leakage, but cannot mitigate privacy risks arising from the

*Table 1.* Notation used in the theoretical analysis.

| Symbol | Definition |
|---|---|
| $K$ | Number of classes |
| $c$ | Class index |
| $x_i,\ X = \{x_i\}_{i=1}^{B}$ | Input of sample $i$ and a mini-batch of inputs |
| $B$ | Mini-batch size |
| $d_\ell, D_\ell$ | Feature and decision-level representation dimension at layer $\ell$ |
| $z_i \in \mathbb{R}^{d_\ell}$ | Representation (feature) of sample $i$ at the current layer $\ell$ |
| $Z = (z^{(1)}, \ldots, z^{(B)}) \in \mathbb{R}^d$ | Batch concatenation of $B$ representations |
| $d = B \cdot d_\ell$ | Dimension of the concatenated batch vector $Z$ |
| $y_i \in \{0,1\}^K$ | One-hot ground-truth label of sample $i$ (so $y_i = e_c$ for class $c$) |
| $\mathcal{I}_c = \{i : y_i = e_c\},\ N_c = |\mathcal{I}_c|$ | Index set and number of samples in class $c$ |
| $\mu_c = \frac{1}{N_c} \sum_{i \in \mathcal{I}_c} z_i$ | Class center |
| $\Sigma_c = \frac{1}{N_c} \sum_{i \in \mathcal{I}_c} (z_i - \mu_c)(z_i - \mu_c)^\top$ | Class covariance |
| $o_i = f_{\text{cloud}}(z_i) \in \mathbb{R}^K$ | Logits for sample $i$ |
| $p_i = \text{softmax}(o_i) \in \Delta^{K-1}$ | Predicted class probabilities |
| $J_i = \frac{\partial o_i}{\partial z_i} \in \mathbb{R}^{K \times d_\ell}$ | Jacobian of logits w.r.t. representation |
| $e_k \in \mathbb{R}^K$ | $k$-th standard basis vector |
| $\delta_i = \frac{\partial \mathcal{L}_{CE}}{\partial o_i} = p_i - y_i$ | CE gradient w.r.t. logits |
| $y_i'$ | Label-smoothed target (A): $y_{i,c}' = 1 - \alpha,\ y_{i,k}' = \alpha/(K-1)\ (k \neq c)$ |
| $\tilde{\delta}_i = \frac{\partial \mathcal{L}_{LS}}{\partial o_i} = p_i - y_i'$ | LS gradient w.r.t. logits |

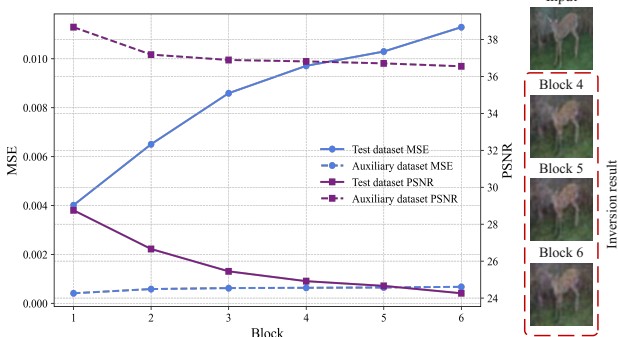

*Figure 2.* Comparison of MSE and PSNR after performing MIA on different depths of ViT (same settings as Sec. 4).

semantic information embedded in $z$. Thus, the threat lies in *what* is transmitted rather than *how* it is transmitted, and encryption alone does not guarantee privacy in CI.

### 2.2. Model Inversion Attack

The goal of the adversary is to reconstruct the original input $x$ from the intermediate information $z$, by learning an approximate inverse mapping $g \approx f_{\text{edge}}^{-1}$. The attack remains feasible under both white-box and black-box conditions: with white-box access the adversary can build a surrogate that more closely approximates the true inverse mapping, while with only black-box access the procedure of (Yang et al., 2019; He et al., 2019; Yin et al., 2023) still enables effective training of the inverse model. The inversion model construction and training details are provided in Section 4.

## 3. Theoretical Analysis for Model Partitioning

This section develops a theory for intrinsic inversion resistance in CI. We show that depth is not the driver: it is sufficient only insofar as it enables a representational transition, while the transition is the necessary mechanism that induces an abrupt increase in the lower bound of $H(X \mid Z)$. We then relate the bound to information dimensionality $D$ and $R_c^2$, yielding an $R_c^2$-guided criterion to locate the GPZ. Finally, we characterize the training dynamics of $R_c^2$ and show that label-distribution design (e.g., label smoothing) can deliberately steer it. Table 1 summarizes the notation used throughout our derivations.

### 3.1. Golden Partition Zone

The mutual information between the input $X$ and intermediate information $Z$ is defined as:

$$I(X;Z) = H(X) - H(X|Z) = H(Z) - H(Z|X) \quad (1)$$

Throughout, $X$ and $Z$ are represented with finite-precision variables, and $H(\cdot)$ denotes Shannon entropy. When the

mapping $f_{\text{edge}} : X \mapsto Z$ is deterministic, as in standard feedforward networks (Xia et al., 2025), $H(Z|X) = 0$, and thus $I(X;Z) = H(Z)$.

In standard forward networks, information transformations form a Markov chain, and by the data processing inequality:

$$I(X; Z_{\ell 1}) \geq I(X; Z_{\ell 2}) \geq \cdots \geq I(X; Z_\ell) \quad (2)$$

implying that $I(X;Z)$ decreases with depth, corresponding to increased $H(X \mid Z)$ and reduced $H(Z)$, which hinders MIA success (Liu et al., 2025).

However, the rate at which $I(X;Z)$ decreases is highly *architecture-dependent*. For example, ResNet-style residual, skip connections and concatenation do not violate Eq. 2, but make layer mappings closer to injective (approximately invertible), thereby slowing the decay of $I(X;Z)$ with depth.

Building on this observation, we present a concrete case study showing that depth alone does not guarantee MIA resistance in ViT. As shown in Fig. 2, even attacking the 6th (final) layer of a ViT (Dosovitskiy et al., 2021) yields accurate reconstructions. Each layer retains a hybrid representation (a 1D decision token plus 256D patch embeddings), which remains rich in instance-specific information. Consequently, increasing depth does not substantially reduce $I(X;Z)$ or induce a representational transition. The model architecture and training code are available on GitHub[1].

Consistently, our ResNet results (Sec. 4.1 and Appendices) show that increasing depth within a certain range does not necessarily hinder MIA and can even facilitate it. One might attribute these cases to the dimensionality of $z$. However, our VGG19 results show the opposite trend even at matched representation dimensionality (Sec. 4.1). Taken together, these results suggest the following empirical takeaway: ***depth is sufficient only insofar as it enables a representational transition, whereas the transition itself is necessary for intrinsic resistance***. This motivates three questions:

---

[1]https://github.com/GoldenPartitionZone/
GoldenPartitionZone

*why does the transition impede MIA, how strong can the impediment be, and can it be deliberately controlled?* We next address them via an information-theoretic analysis.

**Definition 3.1** (Differential Entropy (Shannon, 1948))**.** Let $X$ be a continuous random vector in $\mathbb{R}^d$ with probability density function $f_X(x)$. Its *differential entropy* is defined as

$$h(X) = -\int_{\mathbb{R}^d} f_X(x) \ln f_X(x) \, \mathrm{d}x,$$

provided the integral exists.

**Lemma 3.2** (Maximum Entropy Principle for Fixed Covariance (Jaynes, 1957))**.** *Among all continuous distributions on $\mathbb{R}^d$ with a given covariance matrix $\Sigma$, the Gaussian distribution $\mathcal{N}(\mu, \Sigma)$ uniquely maximizes the differential entropy. In particular, for any $X$ with $\mathrm{Cov}(X) = \Sigma$,*

$$h(X) \leq h\big(\mathcal{N}(\mu, \Sigma)\big) = \tfrac{1}{2}\ln[(2\pi e)^d \det \Sigma].$$

In the following analysis, we first treat intermediate representations as continuous variables and use Definition 3.1 and Lemma 3.2 to upper-bound their differential entropy. We then bridge to the finite-precision setting by incorporating a discretization correction term, yielding corresponding bounds for Shannon entropy.

First, we turn to the feature-level representations. Let $Z_{\text{feat}} \in \mathbb{R}^d$ denote the continuous vector of intermediate features at some layer. By Lemma 3.2, among all distributions on $\mathbb{R}^d$ that share the covariance $\Sigma_{\text{feat}}$, the Gaussian attains the largest differential entropy. Hence

$$h\big(Z_{\text{feat}}\big) \leq h\big(\mathcal{N}(\mu_{feat}, \Sigma_{\text{feat}})\big) = \frac{1}{2}\ln[(2\pi e)^d \det \Sigma_{\text{feat}}]. \tag{3}$$

Moreover, if we let $\sigma_{\text{feat}}^2 = \frac{1}{d}\mathrm{tr}\big(\Sigma_{\text{feat}}\big)$, then by the determinant–trace inequality (Horn & Johnson, 2012) $\det \Sigma_{\text{feat}} \leq (\sigma_{\text{feat}}^2)^d$ it follows that

$$h\big(Z_{\text{feat}}\big) \leq \frac{d}{2}\ln\big(2\pi e\, \sigma_{\text{feat}}^2\big) = O\big(d\ln \sigma_{\text{feat}}^2\big). \tag{4}$$

Next, non-generative neural networks or language models consistently exhibit a clustering behaviour, whereby intermediate representations collapse into the "decision" that encodes class information. If we let $Z_{\text{dec}} \in \mathbb{R}^D$ denote the decision-level information, then its overall entropy satisfies:

$$h\big(Z_{\text{dec}}\big) \leq \max_c h\big(Z_{\text{dec}} \mid Y = c\big) + \ln K. \tag{5}$$

For the decision-level representation, let $\Sigma_c \triangleq \mathrm{Cov}(Z_{\text{dec}} \mid Y = c)$. By Lemma 3.2, and applying the orthogonal spectral decomposition of $\Sigma_c$ together with the Arithmetic Mean–Geometric Mean Inequality (AM-GM) inequality (Hardy et al., 1952), we obtain

$$h\big(Z_{\text{dec}} \mid Y = c\big) \leq \frac{1}{2}\ln[(2\pi e)^D \det \Sigma_c] \leq \frac{D}{2}\ln\big(2\pi e\, \frac{R_c^2}{D}\big) \tag{6}$$

where $R_c^2$ denotes the intra-class mean-squared radius, i.e., $R_c^2 = \mathrm{tr}\big(\Sigma_c\big) = \frac{1}{N_c}\sum_{i:y_i=c}\|z_i - \mu_c\|^2$. Refer to Appendix A for the above calculation details.

Define the entropy upper-bound surrogates

$$\bar{h}_{\text{feat}} \triangleq \frac{d}{2}\ln\big(2\pi e\, \sigma_{\text{feat}}^2\big),$$
$$\bar{h}_{\text{dec}} \triangleq \max_c \frac{D}{2}\ln\big(2\pi e\, \frac{R_c^2}{D}\big) + \ln K. \tag{7}$$

Combining Eqs. (4) and (6) yields the surrogate gap

$$\Delta\bar{h} \triangleq \bar{h}_{\text{feat}} - \bar{h}_{\text{dec}}$$
$$= \frac{d}{2}\ln \sigma_{\text{feat}}^2 - \max_c \frac{D}{2}\ln\big(\frac{R_c^2}{D}\big) + \frac{d-D}{2}\ln(2\pi e) - \ln K. \tag{8}$$

Thus, the gap depends on $d$, $D$, $\sigma_{\text{feat}}^2$, and $R_c^2$. Typically $d \geq D$ (often only 1–4 × larger), and $\sigma_{\text{feat}}^2$ contains both intra- and inter-class variance, whereas $R_c^2$ reflects only the intra-class spread of class $c$. Hence, the first two logarithmic terms with the constant $\frac{d-D}{2}\ln(2\pi e)$ yield a sizeable positive gap, and the $\ln K$ correction is negligible.

Although the bounds are derived for differential entropy, Appendix B provides a continuous–discrete bridge that relates them to the Shannon entropy of finite-precision activations.

Using the upper bounds in Eqs. (4) and (6) together with the information identity in Eq. (1), we can derive a lower bound on $H(X \mid Z)$:

$$H(X \mid Z) \geq H(X) - H(Z_\Delta)$$
$$\geq \begin{cases} H(X) - \frac{d}{2}\ln\big(2\pi e\, \sigma_{\text{feat}}^2\big) \\ \qquad - \kappa_\Delta(d), & Z = Z_{\text{feat}}, \\ H(X) - \frac{D}{2}\ln\big(2\pi e\, \frac{R_c^2}{D}\big) \\ \qquad - H(Y) - \kappa_\Delta(D), & Z = Z_{\text{dec}}. \end{cases} \tag{9}$$

where $\kappa_\Delta(\cdot)$ is the finite-precision correction induced by the discretizer $Q$ at precision $\Delta$ (defined and derived in Appendix C, based on Appendix B).

Since $H(X)$ remains unchanged along the NN information flow, the lower bound on $H(X \mid Z)$ is dominated by the subtracted term. When $Z$ is a decision-level representation, this term $\frac{D}{2}\ln\big(2\pi e\, R_c^2/D\big)$ is typically smaller than its feature-level counterpart, so the bound on $H(X \mid Z)$ is larger, i.e. there is more residual uncertainty about the input. This indicates that ***the decision-level representation and its transition zone exhibit enhanced resistance***, which we define as the *Golden Partition Zone*.

**Active location.** Although the full lower bound in Eq. (9) is difficult to estimate directly, its key quantity $R_c^2$ can be computed layer by layer from class means and within-class

mean squared distances. Monitoring these layer-wise $R_c^2$ values allows active location of the representational transition. Compared with estimating mutual information (Belghazi et al., 2018; Duan et al., 2023), this approach is much easier and provides a practical means to locate the GPZ across architectures, offering deployment-oriented guidance rather than mere post-hoc observation. Pseudocode for automatic location and experimental validation are given in Appendix D.

During the training phase, since the $D$ is determined by the model architecture and the $K$, which constrains upper bound of $H(Y)$, is fixed and generally negligible unless exceedingly large, the primary factor we can manipulate to influence the lower bound is the $R_c^2$. Accordingly, the next question we ask is: ***how does $R_c^2$ evolve through training, and can we deliberately control it?***

### 3.2. Neural Vortex

With cross-entropy training and one-hot vectors, the back-propagated gradient is $\tilde{g}_i = J_i^\top \delta_i$ with $\delta_i = p_i - y_i$ (see Table 1). The backpropagation update $z_i^+ = z_i - \gamma \tilde{g}_i$, thus induces the first-order gradient-induced change in $R_c^2$:

$$\Delta R_c^2 = -\frac{2\gamma}{N_c} \sum_{i \in c} (z_i - \mu_c)^\top \tilde{g}_i \qquad (10)$$

where the $1/N_c$ factor appears because $R_c^2$ is defined as an average. Substituting $\tilde{g}_i = J_i^\top (p_i - y_i)$ gives

$$\Delta R_c^2(\text{one-hot}) = -\frac{2\gamma}{N_c} \sum_{i \in c} \Big[ (p_{ic}-1) T_{\text{corr},i} + \sum_{k \neq c} p_{ik} T_{k,i} \Big] \qquad (11)$$

where $T_{\text{corr},i} = (z_i - \mu_c)^\top J_i^\top e_c$ and $T_{k,i} = (z_i - \mu_c)^\top J_i^\top e_k$. (The full derivation is provided in Appendix E.)

Rewrite the $T$-terms in Eq. 11 via angles. Let $r_i := z_i - \mu_c$. For the correct class, $T_{\text{corr},i} = r_i^\top J_i^\top e_c = \|r_i\| \|J_i^\top e_c\| \cos\theta_{c,i}$, where $\theta_{c,i}$ is the angle between $r_i$ and $J_i^\top e_c$. Similarly, for each off-class $k \neq c$, $T_{k,i} = r_i^\top J_i^\top e_k = \|r_i\| \|J_i^\top e_k\| \cos\theta_{k,i}$. Hence, $T_{\text{corr},i} < 0$ (resp. $> 0$) corresponds to an obtuse (resp. acute) alignment between the residual direction and the $c$-logit gradient direction; the same holds for $T_{k,i}$.

With one-hot labels, $(p_{ic} - 1) \leq 0$. Therefore, in the contraction regime where the representation is pulled toward the class center, the dominant, probability-weighted contribution typically comes from samples with $T_{\text{corr},i} < 0$ (i.e., $\theta_{c,i} > 90°$), making $(p_{ic}-1)T_{\text{corr},i} \geq 0$ and thus pushing the bracketed term in Eq. 11 to be positive on average. In contrast, the off-class sum $\sum_{k \neq c} p_{ik} T_{k,i}$ is sign-mixed, but it is multiplicatively scaled by $p_{ik}$ and quickly diminishes as training proceeds ($p_{ik} \to 0$), so it cannot dominate except possibly at very early epochs. Consequently, $\Delta R_c^2(\text{one-hot})$

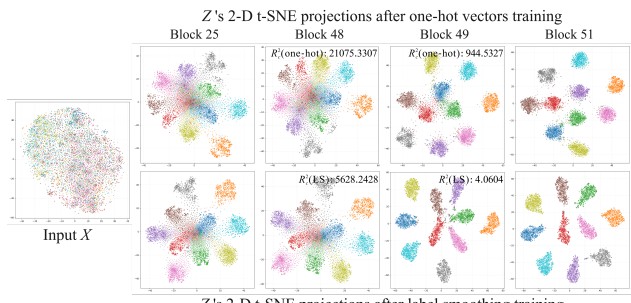

*Z*'s 2-D t-SNE projections after one-hot vectors training

*Z*'s 2-D t-SNE projections after label smoothing training

*Figure 3.* Block-wise 2-D t-SNE of IR-152 information $Z$.

is typically negative during the mid-to-late stage, indicating continued contraction of $R_c^2$, and it gradually vanishes as $p_{ic} \to 1$ and $p_{ik} \to 0$. See Appendix E.4 for the angle interpretation of the $T$-terms.

Under label smoothing (LS) with parameter $\alpha$, the residual becomes $\tilde{\delta}_i = p_i - y_i'$ where $y_c' = 1 - \alpha$ and $y_k' = \alpha/(K-1)$. Thus Eq. 11 carries over by replacing the one-hot coefficients with $(p_{ic} - 1 + \alpha)$ and $(p_{ik} - \alpha/(K-1))$, yielding

$$\begin{aligned} &\Delta R_c^2(\text{LS}) \\ &= -\frac{2\gamma}{N_c} \sum_{i \in c} \Big[ (p_{ic}-1+\alpha) T_{\text{corr},i} + \sum_{k \neq c} \big(p_{ik} - \tfrac{\alpha}{K-1}\big) T_{k,i} \Big] \end{aligned} \qquad (12)$$

Once $p_{ic} > 1 - \alpha$, $(p_{ic} - 1 + \alpha)$ flips sign and, in practice, the alignment of $T_{\text{corr},i}$ reverses, keeping $\Delta R_c^2 < 0$ (see Appendix E.4). Thus, a key property of LS is that it discourages over-confidence: when $p_{ic}$ sharpens beyond the softened target, sustaining a geometric drive that vanishes only as $p_i \to y_i'$. See Appendix F for an explicit lower bound on the gradient norm in this over-confident regime. This correction tends to further reduce $R_c^2$ at the decision-representation layers, which by Eq. (9) increases the lower bound on $H(X \,|\, Z)$ and strengthens resistance to MIA.

In summary, with $T$-term signs locally consistent, $(p_i - y_i)$ is the key lever that reweights the $T$-terms and thereby modulates the direction and magnitude of $\Delta R_c^2$. ***Accordingly, the target label distribution sets this lever, shaping the evolution of $R_c^2$ and the gradient-norm lower bound.***

Different from information-bottleneck methods that explicitly penalize information (Wang et al., 2021) and from neural collapse (Papyan et al., 2020) that describes end-stage geometry a posteriori, we take a training-dynamics perspective and show that reshaping the label distribution, thereby increasing the output entropy, can continuously drive the contraction of $R_c^2$, raise the lower bound of $H(X \,|\, Z)$, and consequently suppress $I(X; Z)$ and $H(Z)$. We refer to this counterintuitive phenomenon, higher output entropy coupled with lower intermediate entropy, as the *Neural Vortex*, which is intended as an analysis-backed explanatory con-

*Table 2.* Dataset, model and intermediate-layer dimension. $^+$ indicates positive factor training. Test accuracy is the maximum achieved after the model fully fits the training set.

| Data splits and auxiliary data | | |
| --- | --- | --- |
| Task | Train / Test | Auxiliary Data |
| CIFAR-10 (10 classes) | 66.6% / 16.7% | 16.7% of CIFAR-10 |
| FaceScrub (530 classes) | 80% / 20% | CelebA (non-individual overlapping) |
| KMNIST (49 classes) | 70% / 15% | 15% of KMNIST, MNIST, EMNIST |
| **Model allocation and accuracy** | | |
| Task | Model | Accuracy (LS$^+$'s $\alpha$ = 0.3, LS$^-$'s $\alpha$ = -0.05) |
| CIFAR-10 | IR-152 | One-hot: 90.59%, LS$^+$: 90.76%, LS$^-$: 90.49% |
| | VGG19 | One-hot: 91.50%, LS$^+$: 91.85% |
| | ViT (6-depth) | One-hot: 78.65%, LS$^+$: 79.26% |
| FaceScrub | VGG19 | LS$^+$: 90.14% |
| KMNIST | VGG19 | One-hot: 97.01% |
| **Dimensions of information at different model depths** | | |
| Model | Block (Depth) | Dimensions |
| IR-152 | 3; 8; 25, 40, 48; 49, 50, 51 | (64,32,32); (128,16,16); (256,8,8); (512,4,4) |
| VGG19 | 13; 17, 26; 30, 39; 43, 52 | (128,32,32); (256,16,16); (512,8,8); (512,4,4) |
| ViT | 1–6 | (257,128) |

cept for the training dynamics discussed above rather than a standalone formal mechanism; an intuitive interpretation is provided in Appendix G.

Figure 3 shows 2-D t-SNE (Van der Maaten & Hinton, 2008) maps of IR-152 (He et al., 2016) $z$ from different blocks after one-hot (top) and label-smoothed (bottom) training. Although t-SNE cannot quantify and reflect entropy or the class radius, it can illustrate that LS tightens class clusters earlier, and the decision layer is much more compact, hinting at a smaller intra-class radius. We directly computed a batch of $z$'s $R_c^2$ at each block and found that the LS model yields a noticeably lower value around the GPZ.

# 4. Experiments

This section describes the experimental setup and analyzes four aspects: representation effects on MIA, intermediate entropy enhancement, inversion-model enhancement, and data-type impacts.

## 4.1. Experiment Setup

*1) Datasets.* We use seven image datasets, all with a resolution of $64 \times 64$. The data allocation is detailed in Table 2, and dataset descriptions are provided in the Appendix H.1.

*2) Target Models.* To examine representational transitions across network depths, we selected four representative vision models: IR-152, ResNet-50 (He et al., 2016), a residual network that mitigates gradient vanishing via skip connections; VGG19 (Simonyan & Zisserman, 2014), a classical sequential convolutional architecture without residuals; and ViT (Dosovitskiy et al., 2021), here using a six-layer configuration (patch size 4, embedding 128, 10 heads) with self-attention to capture global context. Model accuracies on different datasets and the corresponding information dimensions across depths are shown in Table 2.

*3) Inversion model and Implementation details.* We adopt

an inversion architecture based on (Yang et al., 2019; Zhang et al., 2023) and train both target and inversion models with standard optimization settings. To avoid relying on a single fixed attack, we further strengthen the inversion evaluation from two directions as analysis-guided stress tests: on the representation side, we enrich the transmitted information through FFT-based, normalization-based, and NN-based augmentation strategies, with details deferred to Appendix H.4; on the model side, we progressively enhance the inversion architecture through a series of attention mechanisms, as well as an independent reverse-structured residual variant, with details provided in Appendix H.3 and Sec. 4.2 (Part 3). All training hyperparameters and hardware details are deferred to Appendix H.3 & H.4.

*4) Evaluation Metrics.* For evaluation, we use **mean squared error (MSE)** and **peak signal-to-noise ratio (PSNR)** to measure pixel-level reconstruction quality, **structural similarity index measure (SSIM)** to assess structural fidelity, and **learned perceptual image patch similarity (LPIPS)** to evaluate perceptual similarity. We also perform visual comparisons for qualitative analysis. Lower MSE and LPIPS, together with higher PSNR and SSIM, indicate better reconstruction and thus weaker inversion resistance. Compared with prior works that mainly rely on pixel-wise metrics, we include SSIM and especially LPIPS as complementary structural and perceptual metrics. In particular, LPIPS has been shown to correlate better with human perceptual judgments than traditional pixel-wise metrics (Zhang et al., 2018); in our implementation, LPIPS is computed using the default LPIPS setting with an AlexNet backbone (Krizhevsky et al., 2012). Based on our results, an **MSE below 0.02** serves as an empirical threshold for high reconstruction quality, as images at this error level typically exhibit high visual fidelity. The reported results are obtained from attacks on models trained both with and without LS.

## 4.2. Experimental Results

*1) The effect of information representation on MIA.* As shown in Fig. 4, for IR-152, which employs residual skip connections, the forward accumulation of intermediate information slows down the transition of representation. As a result, even up to the 48th block, the model maintains an MSE below 0.02 (an empirical threshold indicating that high-fidelity reconstructions remain possible). However, once reaching Block 49, where the spatial resolution is reduced to $4 \times 4$, the representation abruptly shifts to decision-level, leading to a sharp rise in MSE. This phenomenon is common in residual NNs, where even depths as large as 48 blocks fail to resist MIA effectively, highlighting the necessity of the representational transition. We further observe that dimensionality affects inversion difficulty through the roles of $d$ and $D$ in Eq. (9); however, once the representation undergoes the transition toward the decision-level zone, the

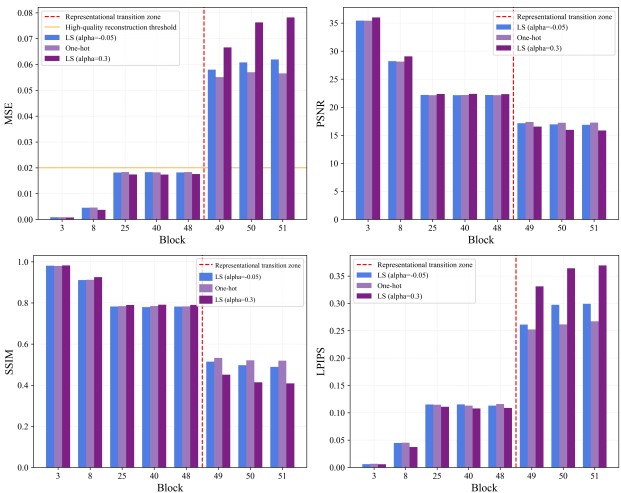

Figure 4. MIA performance results of different IR-152 Blocks.

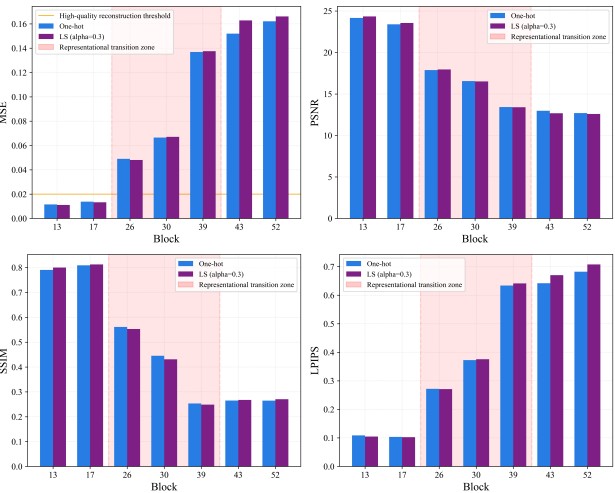

Figure 5. MIA performance results of different VGG19 Depths.

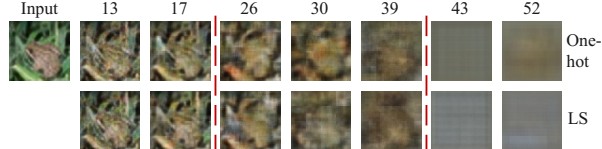

Figure 6. Visual performance of different VGG19 Depths. Dotted lines distinguish the representational transition zone.

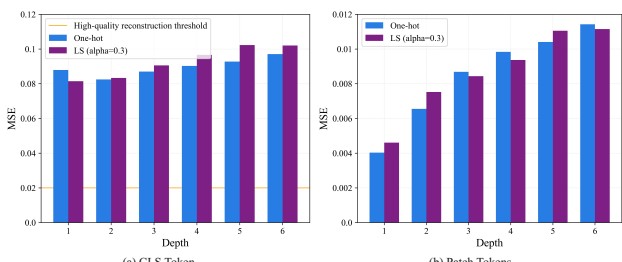

Figure 7. MIA performance results of different ViT Depths.

sharp change in $R_c^2$ becomes more prominent, making the effect of dimensionality relatively less pronounced than in the feature-representation zone. See Appendix H.5 for a targeted analysis of this effect.

It is important to note that the red zones in the figure are based on the rationale established in Section 3. Specifically, before entering the decision zone, LS lacks sufficient strength to contract $R_c^2$. Therefore, we designate the transition zone as the region just before clear MSE divergence emerges between one-hot and LS training.

As shown in Fig. 5, VGG19, lacking residual connections, exhibits a smoother transition in information representation. At Block 39, the MSE rises sharply from 0.066493 at Block 30 to 0.136896, even though the spatial resolution remains at $8 \times 8$, unlike IR-152. While prior studies often assume that decision-level information emerges in the

final linear layers, our results suggest that, in models like VGG, this transition occurs much earlier, in the intermediate blocks. Compared to IR-152, VGG is therefore more suitable for lightweight edge-side partitioning in collaborative inference. As further visualized in Fig. 6, reconstructions before Block 26 remain visually accurate, with MSE consistently below 0.02. Once the model enters the representational transition zone, reconstructions rapidly degrade, and when the information becomes purely decision-level, they lose semantic interpretability altogether.

Finally, as shown in Fig. 7, when attacking using only the CLS token, the behaviour under LS aligns with our theoretical analysis. However, when using patch tokens, the reconstruction quality exhibits fluctuations. This is attributed to the alternating influence of self-attention: one layer may pull representations toward the CLS token (favouring decision-oriented), while the next may refocus on patch-level features, resulting in the observed oscillation.

Notably, ViT's output structure, a fixed number of 256 patch tokens, remains consistent across depths. This architectural design prevents any representational transition; as analyzed in Section 3.1, when the output simultaneously carries both decision and feature information, no GPZ emerges, and thus ViT fails to provide effective resistance against MIA regardless of how deep the partition point is set.

*2) The effect of intermediate information entropy enhancement.* Prior work (Liu et al., 2025) has highlighted the critical role of entropy $H(z)$ in determining MIA susceptibility. To further assess the resistance of the decision-representational zone, we design several enhancement strategies, summarized in Table 3, including an FFT-based method (Fast Fourier Transform(Cooley & Tukey, 1965)); global normalization via mean and variance, and NN-based

*Table 3.* Attack results on Block-50 (decision-level) and Block-40 (feature-level) of IR-152 using different feature enhancement methods. Methods marked with * use the same technique as above.

| Method | Auxiliary Data Set | | Test Data Set | |
|---|---|---|---|---|
| | MSE | PSNR | MSE | PSNR |
| Block-50-baseline | 0.021366 | 21.500321 | 0.057010 | 17.224045 |
| **Augment by Analysis** | | | | |
| FFT (Residual) | 0.089661 | 15.271254 | 0.112854 | 14.260188 |
| FFT (Concat) | **0.017355** | **22.411592** | 0.068011 | 16.458754 |
| Normalize (Input) | 0.022920 | 21.195560 | 0.056266 | 17.283856 |
| Normalize (Residual) | 0.022929 | 21.192309 | 0.056317 | 17.279251 |
| Normalize (Concat) | 0.023193 | 21.145331 | **0.055434** | **17.344250** |
| **Augment by NN** | | | | |
| w/o-Dropout (Residual) | **0.014784** | **23.099889** | 0.061043 | 16.929223 |
| with-Dropout (Residual) | 0.016584 | 22.591667 | 0.055617 | 17.334325 |
| w/o-Dropout (Concat) | 0.018930 | 22.038503 | 0.057163 | 17.213750 |
| with-Dropout (Concat) | 0.016095 | 22.722609 | **0.051740** | **17.646756** |
| **Augment by Analysis & NN (Concat)** | | | | |
| Normalize (Input) & w/o-Dropout | **0.015285** | **22.958731** | 0.055669 | 17.326721 |
| Normalize (Input) & with-Dropout* | 0.015489 | 22.891474 | **0.051937** | **17.627761** |
| Block-40-baseline | 0.011796 | 24.080942 | 0.018245 | 22.171516 |
| Block-40-Augment* | **0.004460** | **28.299952** | **0.014026** | **23.318254** |

*Table 4.* Attack results on Block-50 and Block-40 of IR-152 using different inversion model enhancement mechanisms.

| Method | Auxiliary Data Set | | Test Data Set | |
|---|---|---|---|---|
| | MSE | PSNR | MSE | PSNR |
| Block-50-baseline | 0.021366 | 21.500321 | 0.057010 | 17.224045 |
| MHA | 0.018156 | 22.200025 | 0.052159 | 17.610650 |
| AttentionAsConv | 0.020217 | 21.736728 | 0.053299 | 17.515945 |
| AttentionAsConv+SE | 0.017121 | 22.458065 | 0.051737 | 17.647012 |
| AttentionAsConv+SE +LSK+MSCA | 0.019283 | 21.942643 | 0.050579 | 17.743654 |
| Attention&Augment by NN with Dropout (Concat)* | 0.013986 | 23.333684 | **0.049272** | **17.861294** |
| Inversion-IR152 | **0.013218** | **23.589063** | 0.051795 | 17.643729 |
| Block-40-baseline | 0.011796 | 24.080942 | 0.018245 | 22.171516 |
| Block-40-Augment* | **0.003258** | **29.662543** | **0.011905** | **24.036657** |

enhancement. $z$ is either combined residually with, or concatenated with, its enhanced counterpart. See more details in Appendix H.4. Our experiments show that improper enhancement can cause the inversion model to overfit, i.e., achieve strong reconstruction on the auxiliary set while lacking generalization. NN-based methods without dropout are particularly prone to this. In contrast, combining global normalization with dropout-regularized NN modules effectively mitigates overfitting and improves attack quality. For example, PSNR gains on IR-152 reach only +0.4 at Block 50 but rise to around +1.13 at Block 40, indicating that enhancement strategies are more effective in the feature representation zone, while the decision-level zone remains robust against MIA.

*3) The effect of inversion model enhancement.* The design of the inversion model plays a critical role in capturing the true conditional entropy $H(X \mid Z)$. Based on existing architecture, we introduce attention mechanisms between deconvolutional blocks to improve representational capacity and mitigate overfitting. We first experimented with multi-head attention (MHA) (Vaswani et al., 2017), which led to performance gains but incurred nearly 10× training

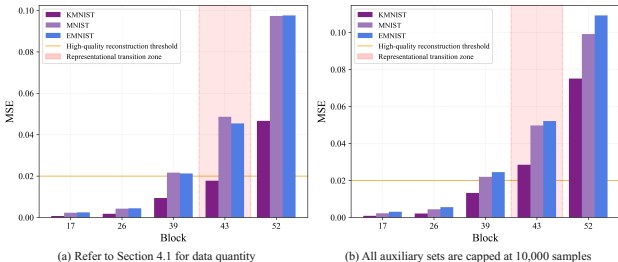

(a) Refer to Section 4.1 for data quantity  (b) All auxiliary sets are capped at 10,000 samples

*Figure 8.* Reconstruction performance of KMNIST data under different auxiliary data sets of VGG19.

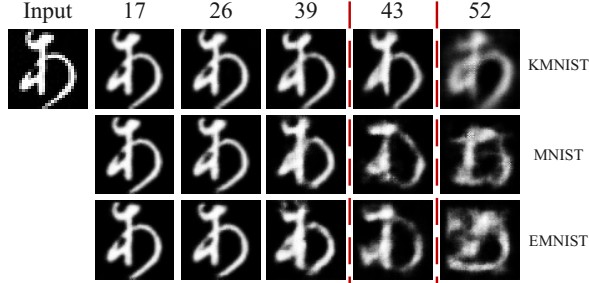

*Figure 9.* Visual reconstruction performance of KMNIST data under different auxiliary data sets of VGG19.

cost. To balance efficiency and performance, we adopted the lightweight Attention-as-Conv. To further enhance the model, we incrementally integrated this into shallow blocks, augmented with Squeeze-and-Excitation (SE) modules (Hu et al., 2018). In deeper blocks, we introduced large kernel attention (LSK) (Peng et al., 2017) and multiscale channel attention (MSCA) (Liu et al., 2021) for stronger decoupling. Importantly, attention design in the inversion model should follow a weak-to-strong decoupling principle. Light attention in shallow layers helps avoid overfitting, while deeper decoupled outputs improve the expressiveness of $H(X \mid Z)$. Our analysis confirms this intuition. Combined with $H(z)$ enhancement, the improved model raises PSNR by +0.64 at Block 50 of IR-152. However, compared to the +1.9 gain in the feature representation zone, this still reflects the inherently stronger MIA resistance of decision-level representations. Additionally, we explore a reverse-structured variant of the IR-152 residual block in the inversion model, which shows stronger expressiveness than basic deconvolutional stacks, though it still fails to achieve effective inversion in the decision zone.

*4) The role of data types.* We conduct experiments using three greyscale datasets, targeting a handwritten Japanese character recognizer and using different auxiliary data types: KMNIST, MNIST and EMNIST. As shown in Fig. 8, due to the simplicity of the input data, the representational transition zone is delayed. The presence of many zero-valued pixels in KMNIST causes feature extraction to persist deeper into the network, with the GPZ occurring around Block 43.

When training the inversion model with different auxiliary datasets, we observe that data from the same distribution yield significantly better reconstruction than similar but out-of-distribution datasets. Among the latter, performance improves with greater distributional proximity to the target. We also find that the larger sample size of EMNIST improves attack quality, but when sample sizes are equalized, MNIST performs better, suggesting it is closer in distribution to KMNIST. As visualized in Fig. 9, reconstructions before the transition zone are accurate across all auxiliary datasets. However, after entering the GPZ, the output drifts toward the auxiliary data distribution: MNIST-trained models tend to produce "0"-like digits, while EMNIST-trained models resemble characters like "D". See Appendix H.7 for more analysis.

## 5. Discussion

*1) Practical deployment guidance.*   For practical privacy-preserving CI, partition selection should balance intrinsic MIA resistance against deployment costs, including edge latency, communication and memory overheads, and energy. Based on Sec. 4.2, Appendices H–I, VGG-like backbones provide the most favorable trade-off among the evaluated architectures: they reach a GPZ with only 2.5% edge-side parameters, while ResNet-style models delay the transition and would require over 78% of parameters on the edge. More importantly, Appendix I shows that within VGG, moving from depth-26 to depth-30 nearly preserves edge latency while halving the transmitted payload, offering a practical split knob under bandwidth constraints; by contrast, although the deep IR152 split further reduces transmission size, it incurs much larger edge compute, model footprint, and energy cost. Taken together, these results provide lightweight deployment guidance for selecting architectures and partition points with stronger intrinsic resistance to MIA under real resource constraints.

*2) Architecture specificity.*   Based on our results and analysis, the emergence of a GPZ is closely related to architectural design. Common edge-model designs such as residual connections and Transformer architectures with persistent patch representations tend to retain feature information across depth and thereby attenuate or delay the representational transition, which in turn benefits MIA. We therefore reiterate that increasing depth alone does not necessarily make MIA harder (Ding et al., 2024).

*3) Data dependence.*   Based on our results and analysis, the emergence and location of the GPZ are also data-dependent. Across the datasets studied in this work, different data characteristics lead to different transition patterns: CIFAR-10 shows a more gradual transition and a wider GPZ (Fig. 6), FaceScrub exhibits an earlier and narrower transition (Fig. 11), while MNIST and KMNIST shift the GPZ deeper and make it narrower because feature extraction persists across more layers (Fig. 9). This observation is consistent with Eq. (9), where inversion difficulty is jointly affected by the data entropy term and $R_c^2$. Under more adverse user distributions, a smaller $H(X)$ and a larger $R_c^2$ at the candidate split layer would weaken the lower bound on $H(X \mid Z)$, delay the onset of intrinsic resistance, and potentially require deeper splitting before a GPZ appears. At the same time, such worst-case-like distributions are less representative of practical CI deployment, which is typically motivated by more complex visual data rather than low-variance, sparsely structured inputs such as MNIST. In this sense, realistic CI data are more likely to resemble complex natural-image settings such as FaceScrub than the more adverse low-variance cases discussed above. A fuller analysis of such worst-case-like distributions is left for future work.

*4) Scope and possible extension.*   The current study focuses on images and deep vision models, so we are cautious about making strong claims regarding the text domain without direct empirical evidence. Nevertheless, our results on ViT suggest a cautious hypothesis for Transformer-style models: when a fixed set of patch tokens persists across depth and continues to preserve sample-level information, the representational transition may be attenuated, making a clear GPZ harder to form. In this sense, whether a GPZ can emerge may depend less on depth itself than on whether deeper hidden states continue to retain token-level information. Verifying this hypothesis in text Transformers and other sequence models is a direction for future work.

## 6. Conclusion

In this paper, we revisit model inversion threats in CI and show that depth alone is not a reliable driver of resistance; it is sufficient only insofar as it enables a representational transition. At this transition, the lower bound of $H(X \mid Z)$ rises abruptly, and the governing variance term shifts from global total variance to the intra-class mean-squared radius $R_c^2$, motivating an $R_c^2$-based rule to locate (or post hoc identify) the *Golden Partition Zone*. We further show that $R_c^2$ follows training dynamics that can be controlled via the label distribution, which we refer to as the *Neural Vortex*, an explanatory concept grounded in our analysis. Our experiments on representative vision models reveal that partitioning at the decision-level representations yields an average 4× increase in MSE compared to feature-level representations, and maintains a 66% advantage in resistance even when intermediate information entropy and inversion models are both enhanced. Additionally, the transition boundary depends on both data type and class count, and larger intra-class variance tends to increase reconstruction resistance.

## Acknowledgements

The authors would like to thank the anonymous reviewers for their valuable comments and constructive suggestions. This work was supported in part by the National Natural Science Foundation of China under Grant No. U2433205 and the Postgraduate Research and Practice Innovation Program of Jiangsu Province under Grant No. KYCX24_0595.

## Impact Statement

This paper presents work whose goal is to advance the field of Machine Learning and Collaborative Inference (CI). As edge deployment becomes increasingly common, protecting CI pipelines against model inversion attacks often relies on add-on defences that introduce privacy–utility trade-offs and extra computation. We instead provide principled guidance for privacy-preserving model partitioning by identifying where representations undergo a transition that sharply increases reconstruction uncertainty. By clarifying this training-dynamics mechanism, our findings can support CI deployments that are more robust to inversion without relying solely on heavy perturbation. We anticipate that this can support more privacy-preserving CI deployments.

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

## Appendix Roadmap

This roadmap is provided because the appendices are divided into multiple largely self-contained sections. It is intended to help readers quickly locate proofs, derivations, algorithms, and supporting experimental details. For each item below, the boldface parenthetical tag indicates the main-text section where the appendix is primarily used.

- **Appendix A. Proofs of entropy upper bounds (Sec. 3.1).** Provides full proofs for the entropy upper bounds used in the main theory, including the feature-level bound and the class-conditional bound (supporting the bound used in Eq. (6)).

- **Appendix B. From differential entropy to discrete Shannon entropy under finite precision (Sec. 3.1).** Derives the bridge from differential entropy to discrete Shannon entropy under finite-precision or quantized activations, justifying the use of discrete entropy quantities in the analysis.

- **Appendix C. Derivation of Eq. (9) (Sec. 3.1).** Gives the detailed derivation of Eq. (9), and explains how the finite-precision quantization induces the resolution-dependent term (e.g., $\kappa_\Delta$) in the bound.

- **Appendix D. Pseudocode for Active Location of the Golden Partition Zone (Sec. 3.1).** Provides pseudocode and implementation details for automatic GPZ location and its experimental validation.

- **Appendix E. First-order change of $R_c^2$ under one-hot and label smoothing (Sec. 3.2).** Gives the complete derivation for the first-order change of $R_c^2$ (e.g., the derivation behind Eq. (11)), and extends the same analysis to label smoothing.

- **Appendix F. Gradient-norm lower bound under label smoothing (Sec. 3.2).** Provides the explicit lower bound on the feature-gradient norm referenced in the main text (e.g., Eq. (53)), supporting the claim that label smoothing can sustain representation contraction effects.

- **Appendix G. Intuitive interpretation for the *Neural Vortex* metaphor (Sec. 3.2).** Contains an intuitive interpretation supporting the *Neural Vortex* metaphor, clarifying how output-entropy effects and representation geometry relate.

- **Appendix H. Supplement for Experiment (Sec. 4).** Collects experimental details: datasets, target models, inversion models, implementation, hyperparameters, hardware, plus additional analyses, ablations (e.g., residual networks, VGG, label-smoothing reconstruction behavior, and data-type studies).

- **Appendix I. Deployment overhead (Sec. 5).** Provides a detailed deployment-oriented cost assessment supporting the practical CI guidance, including edge-side parameter fraction and overhead considerations.

- **Appendix J. Related Work.** Summarizes prior model inversion attacks and defences in collaborative inference, and positions our paper.

## A. Proofs of entropy upper bounds

This appendix provides detailed derivations for the entropy upper bounds used in Sec. 3. Throughout, $\ln$ denotes the natural logarithm. We use $h(\cdot)$ for differential entropy of continuous variables and $H(\cdot)$ for Shannon entropy of discrete variables.

### A.1. Feature-level entropy bound (Eq. (4))

**Setup.** Let $Z_{\text{feat}} \in \mathbb{R}^d$ be a continuous random vector representing feature-level information at a given layer, with mean $\mu_{\text{feat}} = \mathbb{E}[Z_{\text{feat}}]$ and covariance

$$\Sigma_{\text{feat}} \;=\; \text{Cov}(Z_{\text{feat}}) \;=\; \mathbb{E}\big[(Z_{\text{feat}} - \mu_{\text{feat}})(Z_{\text{feat}} - \mu_{\text{feat}})^\top\big].$$

Define the average per-dimension variance

$$\sigma_{\text{feat}}^2 \;\triangleq\; \frac{1}{d}\,\text{tr}(\Sigma_{\text{feat}}).$$

**Lemma A.1** (Maximum-entropy upper bound with fixed covariance). *Among all continuous distributions on $\mathbb{R}^d$ with covariance $\Sigma_{\text{feat}}$, the Gaussian $\mathcal{N}(\mu_{\text{feat}}, \Sigma_{\text{feat}})$ maximizes differential entropy. In particular,*

$$h(Z_{\text{feat}}) \;\leq\; h\big(\mathcal{N}(\mu_{\text{feat}}, \Sigma_{\text{feat}})\big) \;=\; \frac{1}{2}\ln\big[(2\pi e)^d \det \Sigma_{\text{feat}}\big].$$

*Proof.* This follows directly from the maximum entropy principle for fixed covariance (Lemma 3.2 in the main text). For completeness, the Gaussian with covariance $\Sigma_{\text{feat}}$ uniquely maximizes $h(\cdot)$, and its differential entropy admits the closed form $\frac{1}{2}\ln\big[(2\pi e)^d \det \Sigma_{\text{feat}}\big]$ when $\Sigma_{\text{feat}}$ is positive definite. If $\Sigma_{\text{feat}}$ is singular, the Gaussian is degenerate and the differential entropy is $-\infty$, making the inequality trivial. $\square$

**Lemma A.2** (Determinant–trace inequality). *For any* real symmetric *positive semidefinite matrix* $\Sigma \in \mathbb{R}^{d \times d}$,

$$\det(\Sigma) \;\leq\; \left(\frac{\text{tr}(\Sigma)}{d}\right)^d.$$

*Proof.* Since $\Sigma$ is real and symmetric, it admits an orthogonal eigendecomposition $\Sigma = U\Lambda U^\top$, where $U$ is orthogonal and $\Lambda = \text{diag}(\lambda_1, \ldots, \lambda_d)$ with $\lambda_i \in \mathbb{R}$. Moreover, $\Sigma \succeq 0$ implies $\lambda_i \geq 0$ for all $i$ (e.g., by the Rayleigh quotient $u_i^\top \Sigma u_i = \lambda_i \geq 0$ for each eigenvector $u_i$).

Therefore,

$$\det(\Sigma) = \det(\Lambda) = \prod_{i=1}^{d} \lambda_i, \qquad \text{tr}(\Sigma) = \text{tr}(\Lambda) = \sum_{i=1}^{d} \lambda_i.$$

Applying the AM–GM inequality to the nonnegative scalars $\{\lambda_i\}_{i=1}^{d}$ yields

$$\left(\prod_{i=1}^{d} \lambda_i\right)^{1/d} \;\leq\; \frac{1}{d}\sum_{i=1}^{d}\lambda_i,$$

hence

$$\det(\Sigma) = \prod_{i=1}^{d} \lambda_i \;\leq\; \left(\frac{1}{d}\sum_{i=1}^{d}\lambda_i\right)^d = \left(\frac{\text{tr}(\Sigma)}{d}\right)^d.$$

$\square$

**Lemma A.3** (Feature-level entropy upper bound in Eq. (3)). *With* $\sigma_{\text{feat}}^2 = \frac{1}{d}\text{tr}(\Sigma_{\text{feat}})$,

$$h(Z_{\text{feat}}) \;\leq\; \frac{d}{2}\ln\!\Big(2\pi e\,\sigma_{\text{feat}}^2\Big) \;=\; O\!\big(d\ln \sigma_{\text{feat}}^2\big).$$

*Proof.* Start from Lemma A.1:

$$h(Z_{\text{feat}}) \leq \frac{1}{2}\ln\big[(2\pi e)^d \det \Sigma_{\text{feat}}\big].$$

Apply Lemma A.2 with $\Sigma = \Sigma_{\text{feat}}$:

$$\det \Sigma_{\text{feat}} \leq \left(\frac{\text{tr}(\Sigma_{\text{feat}})}{d}\right)^d = (\sigma_{\text{feat}}^2)^d.$$

Substitute this bound into the previous inequality:

$$\begin{aligned}
h(Z_{\text{feat}}) &\leq \frac{1}{2}\ln\big[(2\pi e)^d (\sigma_{\text{feat}}^2)^d\big] = \frac{1}{2}\ln\big[(2\pi e\,\sigma_{\text{feat}}^2)^d\big] \\
&= \frac{d}{2}\ln\!\Big(2\pi e\,\sigma_{\text{feat}}^2\Big),
\end{aligned}$$

which is exactly Eq. (3), (4) in the main text. $\square$

### A.2. Bounding $h(Z_{\text{dec}})$ by class-conditional entropies Eq. (5)

**Setup.** Let $Y \in \{1, \ldots, K\}$ be a discrete class label and $Z_{\text{dec}} \in \mathbb{R}^D$ be a continuous decision-level representation.

**Lemma A.4** (Reduction of $h(Z_{\text{dec}})$ to class-conditional entropies). *The overall decision-level entropy satisfies*

$$h(Z_{\text{dec}}) \;\leq\; \max_c h(Z_{\text{dec}} \mid Y = c) + H(Y) \;\leq\; \max_c h(Z_{\text{dec}} \mid Y = c) + \ln K.$$

*Proof.* **Step 1: Use the identity linking mutual information and conditional differential entropy.** For a discrete $Y$ and continuous $Z_{\text{dec}}$, mutual information can be written as

$$I(Y; Z_{\text{dec}}) = h(Z_{\text{dec}}) - h(Z_{\text{dec}} \mid Y),$$

where

$$h(Z_{\text{dec}} \mid Y) \triangleq \sum_{c=1}^{K} \Pr(Y = c) \, h(Z_{\text{dec}} \mid Y = c)$$

is the conditional differential entropy averaged over $Y$. Rearranging gives

$$h(Z_{\text{dec}}) = h(Z_{\text{dec}} \mid Y) + I(Y; Z_{\text{dec}}).$$

**Step 2: Upper bound $I(Y; Z_{\text{dec}})$ by $H(Y)$.** Using the alternative expression

$$I(Y; Z_{\text{dec}}) = H(Y) - H(Y \mid Z_{\text{dec}}),$$

and the fact that conditional Shannon entropy is nonnegative, $H(Y \mid Z_{\text{dec}}) \geq 0$, we obtain

$$I(Y; Z_{\text{dec}}) \leq H(Y).$$

Moreover, since $Y$ takes $K$ values, $H(Y) \leq \ln K$.

**Step 3: Upper bound $h(Z_{\text{dec}} \mid Y)$ by $\max_c h(Z_{\text{dec}} \mid Y = c)$.** Because $h(Z_{\text{dec}} \mid Y)$ is a convex combination of $\{h(Z_{\text{dec}} \mid Y = c)\}_{c=1}^{K}$,

$$h(Z_{\text{dec}} \mid Y) = \sum_{c=1}^{K} \Pr(Y = c) \, h(Z_{\text{dec}} \mid Y = c) \leq \max_c h(Z_{\text{dec}} \mid Y = c).$$

**Step 4: Combine the bounds.** Putting Steps 1–3 together,

$$h(Z_{\text{dec}}) = h(Z_{\text{dec}} \mid Y) + I(Y; Z_{\text{dec}}) \leq \max_c h(Z_{\text{dec}} \mid Y = c) + H(Y) \leq \max_c h(Z_{\text{dec}} \mid Y = c) + \ln K.$$

$\square$

### A.3. Class-conditional entropy upper bound (Eq. (6))

**Setup and meaning of $R_c^2$.** For each class $c$, let

$$\mu_c \triangleq \mathbb{E}[Z_{\text{dec}} \mid Y = c], \qquad \Sigma_c \triangleq \text{Cov}(Z_{\text{dec}} \mid Y = c).$$

We define the intra-class mean-squared radius as

$$R_c^2 \triangleq \text{tr}(\Sigma_c).$$

Equivalently, $R_c^2$ is the class-conditional mean squared deviation from the class mean:

$$\text{tr}(\Sigma_c) = \text{tr}\Big(\mathbb{E}\big[(Z_{\text{dec}} - \mu_c)(Z_{\text{dec}} - \mu_c)^\top \mid Y = c\big]\Big) = \mathbb{E}\big[\|Z_{\text{dec}} - \mu_c\|_2^2 \mid Y = c\big].$$

With $N_c$ samples from class $c$ and decision-level representations $\{z_i\}$, the empirical estimate is

$$R_c^2 = \frac{1}{N_c} \sum_{i:\, y_i = c} \|z_i - \mu_c\|_2^2, \qquad \mu_c = \frac{1}{N_c} \sum_{i:\, y_i = c} z_i.$$

**Lemma A.5** (Entropy upper bound for class-conditional Gaussian)**.** *For each class $c$,*

$$h(Z_{\text{dec}} \mid Y = c) \leq \frac{D}{2} \ln\left(2\pi e \cdot \frac{R_c^2}{D}\right).$$

*Proof.* **Step 1: Start from the closed-form Gaussian entropy.** If $\Sigma_c$ is positive definite, the multivariate Gaussian differential entropy is

$$h(Z_{\text{dec}} \mid Y = c) = \frac{1}{2} \ln\big[(2\pi e)^D \det \Sigma_c\big].$$

If $\Sigma_c$ is singular, then $\det \Sigma_c = 0$ and the (degenerate) Gaussian differential entropy is $-\infty$, so the desired upper bound holds trivially.[2]

**Step 2: Upper bound** $\det \Sigma_c$ **by its trace.** Apply Lemma A.2 to the $D \times D$ matrix $\Sigma_c$:

$$\det \Sigma_c \leq \left(\frac{\text{tr}(\Sigma_c)}{D}\right)^D = \left(\frac{R_c^2}{D}\right)^D.$$

**Step 3: Substitute and simplify.** Substitute the determinant bound into the Gaussian entropy expression:

$$h(Z_{\text{dec}} \mid Y = c) \leq \frac{1}{2} \ln\left[(2\pi e)^D \left(\frac{R_c^2}{D}\right)^D\right]$$

$$= \frac{1}{2} \ln\left[\left(2\pi e \cdot \frac{R_c^2}{D}\right)^D\right] = \frac{D}{2} \ln\left(2\pi e \cdot \frac{R_c^2}{D}\right),$$

which matches Eq. (6) in the main text. $\square$

## B. From differential entropy to discrete Shannon entropy under finite precision

Our theory is stated for continuous representations and hence uses differential entropy $h(\cdot)$. In actual deployments, however, intermediate activations are stored and transmitted in finite precision (e.g., FP16/FP32 or INT8), i.e., as discrete tensors, for which the natural quantity is the discrete Shannon entropy $H(\cdot)$. This appendix formalizes a standard bridge between the two. The key message is:

Finite-precision storage can be viewed as a discretizer $Q$ that induces a partition $\{C_k\}$ of $\mathbb{R}^m$. The discrete entropy $H(Q(Z))$ equals the continuous entropy $h(Z)$ plus a deterministic "cell-volume" offset and a nonnegative mismatch term (a KL divergence). Uniform quantization with step size $\Delta$ is a special case, but the same bridge also applies to non-uniform discretizations such as floating point.

Throughout, $\log$ denotes the natural logarithm (nats).

**General finite-precision discretization model (covers FP32/INT8).** Let $Z \in \mathbb{R}^m$ be a continuous random vector with pdf $f$ and finite differential entropy

$$h(Z) = -\int_{\mathbb{R}^m} f(z) \log f(z)\, dz.$$

Let $Q$ be a (possibly non-uniform) discretizer mapping $\mathbb{R}^m$ to a countable alphabet. The pre-images of symbols form a measurable partition $\{C_k\}$ of $\mathbb{R}^m$ (quantization cells). Assume the cells have finite volume $\text{vol}(C_k) \in (0, \infty)$ for all $k$ that occur with nonzero probability. Define

$$p_k := \mathbb{P}(Z \in C_k) = \int_{C_k} f(z)\, dz, \qquad H(Q(Z)) := -\sum_k p_k \log p_k.$$

Define the associated piecewise-constant (histogram) density

$$f_Q(z) := \frac{p_k}{\text{vol}(C_k)} \quad \text{for } z \in C_k.$$

Note that $f_Q$ is a valid pdf on $\mathbb{R}^m$ and is constant within each cell.

---

[2]If additionally $R_c^2 = 0$, then the right-hand side becomes $\frac{D}{2} \ln(0) = -\infty$, and the inequality holds with equality.

**Exact bridge identity.**

**Lemma B.1** (Exact entropy bridge under a partition). *With the above definitions,*

$$H(Q(Z)) = h(Z) + \mathbb{E}\left[\log \frac{1}{\text{vol}(C_{Q(Z)})}\right] + D(f \,\|\, f_Q), \tag{13}$$

*where $D(f\|f_Q) = \int f(z) \log \frac{f(z)}{f_Q(z)}\, dz \geq 0$ is the KL divergence. Equivalently, defining the compensated discrete entropy*

$$\widetilde{h}_Q(Z) := H(Q(Z)) + \mathbb{E}\left[\log \text{vol}(C_{Q(Z)})\right], \tag{14}$$

*we have the exact identity*

$$\widetilde{h}_Q(Z) = h(Z) + D(f \,\|\, f_Q) \geq h(Z). \tag{15}$$

*Proof.* For $z \in C_k$, $\log f_Q(z) = \log(p_k/\text{vol}(C_k))$. Therefore,

$$-\int f(z) \log f_Q(z)\, dz = -\sum_k \int_{C_k} f(z) \log\left(\frac{p_k}{\text{vol}(C_k)}\right) dz = -\sum_k p_k \log p_k + \sum_k p_k \log \text{vol}(C_k).$$

The right-hand side is $H(Q(Z)) + \mathbb{E}[\log \text{vol}(C_{Q(Z)})]$, which by definition equals $\widetilde{h}_Q(Z)$. Subtracting $h(Z) = -\int f \log f$ yields

$$\widetilde{h}_Q(Z) - h(Z) = \int f(z) \log \frac{f(z)}{f_Q(z)}\, dz = D(f\|f_Q) \geq 0,$$

which proves (15). Rearranging gives (13). $\qquad\square$

**Interpretation.** Equation (13) decomposes the discrete entropy into three parts: (i) the continuous entropy $h(Z)$; (ii) an offset that depends on the typical cell volume seen under $Z$; and (iii) a nonnegative discretization mismatch term $D(f\|f_Q)$, which is small when $f$ varies slowly within cells.

**Uniform quantization as a special case.** Consider coordinate-wise uniform quantization with step size $\Delta > 0$ on each coordinate. Then every cell has volume $\text{vol}(C_k) = \Delta^m$, hence $\mathbb{E}[\log \text{vol}(C_{Q(Z)})] = m \log \Delta$, and (13) reduces to

$$H(Z_\Delta) = h(Z) + m \log \tfrac{1}{\Delta} + r_Z(\Delta), \qquad r_Z(\Delta) = D(f\|f_\Delta) \geq 0, \tag{16}$$

where $f_\Delta$ is the corresponding histogram density and $Z_\Delta := Q_\Delta(Z)$. This is the standard differential-to-discrete relation under fine quantization (e.g., Cover–Thomas).

**High-resolution limit.** If the partition is refined so that cells become small on the region where $f$ concentrates (e.g., $\Delta \to 0$ in uniform quantization, or increasing floating-point precision), then under mild regularity assumptions one has $D(f\|f_Q) \to 0$, and consequently $\widetilde{h}_Q(Z) \to h(Z)$. This is an asymptotic statement with respect to the fineness of the discretization, not a claim that the runtime hardware precision "tends to zero."

**Special case: neural network features stored in FP32 (non-uniform discretization).** Neural network activations stored in IEEE-754 floating point (e.g., FP32) are not produced by a uniform quantizer with a single global step size $\Delta$. Instead, FP32 induces a non-uniform grid whose spacing depends on the exponent (the unit-in-the-last-place, ULP). Roughly, for values with magnitude $|z| \approx 2^e$, the local spacing is on the order of $2^{e-23}$ due to the 24-bit significand (including the hidden bit), i.e., a relative precision of about $2^{-23}$ around that scale. Therefore, it is generally incorrect to model FP32 storage as uniform quantization with a single $\Delta$ over all of $\mathbb{R}^m$.

Nevertheless, Lemma B.1 applies directly to floating point by treating IEEE-754 rounding as a discretizer $Q$ with non-uniform cells $\{C_k\}$. For strict applicability of (13), we restrict attention to the *finite* representable region (i.e., excluding $\pm\infty$ and NaNs) and assume that overflow/exception events occur with zero (or negligible) probability under the population distribution of activations. Under this mild assumption, every cell with $p_k > 0$ has a finite volume, and the bridge (13) holds exactly, with a distribution-dependent cell-volume term $\mathbb{E}[\log 1/\text{vol}(C_{Q(Z)})]$ and mismatch term $D(f\|f_Q)$.

In typical neural networks, activations are numerically bounded (or effectively bounded after normalization/clipping) and concentrate on moderate magnitudes where FP32 cells are extremely small. In regimes where the induced cell diameters are sufficiently fine on the high-probability region of $Z$, the mismatch term $D(f\|f_Q)$ is expected to be small, and the continuous-entropy theory provides an accurate description of finite-precision measurements.

**Carrying over an entropy contraction across layers.** Let $Z_{\text{feat}} \in \mathbb{R}^m$ and $Z_{\text{dec}} \in \mathbb{R}^M$ be the (feature-level and decision-level) continuous representations considered in the main text, and let $Q$ denote the finite-precision discretizer used at run time (e.g., FP32 rounding, or INT8 quantization). Applying Lemma B.1 to both yields

$$\widetilde{h}_Q(Z_{\text{feat}}) - \widetilde{h}_Q(Z_{\text{dec}}) = \big(h(Z_{\text{feat}}) - h(Z_{\text{dec}})\big) + D(f_{\text{feat}}\|f_{\text{feat},Q}) - D(f_{\text{dec}}\|f_{\text{dec},Q}),$$

where $f_{\text{feat},Q}$ and $f_{\text{dec},Q}$ are the corresponding histogram densities. Hence, any population-level contraction established for differential entropy (e.g., $h(Z_{\text{feat}}) > h(Z_{\text{dec}})$ with a nontrivial margin) persists for sufficiently fine discretizations, since both KL terms vanish as the induced cells become small.

**Remark (raw $H(\cdot)$ vs. compensated $\widetilde{h}_Q(\cdot)$).** When $m \neq M$, comparing raw discrete entropies $H(Q(Z_{\text{feat}}))$ and $H(Q(Z_{\text{dec}}))$ may conflate genuine information contraction with deterministic effects caused by different dimensionalities and cell volumes. The compensated entropy $\widetilde{h}_Q(\cdot)$ removes the cell-volume artifact and aligns the finite-precision measurement with the continuous theory.

## C. Derivation of Eq. (9)

We derive the lower bound on $H(X \mid Z)$ used in Eq. (9) under finite precision. At run time, the transmitted/stored activation is discrete due to finite precision; we denote this runtime representation by $Z_\Delta := Q(Z)$, where $Q$ is the discretizer (e.g., FP32 rounding or INT8 quantization). Accordingly, $H(\cdot)$ in this appendix denotes discrete Shannon entropy, and $H(X \mid Z)$ in Eq. (9) should be understood as $H(X \mid Z_\Delta)$.

**Step 1: A generic lower bound via mutual information.** Using the Shannon-entropy identity,

$$H(X \mid Z_\Delta) = H(X) - I(X; Z_\Delta), \tag{17}$$

and the fact that for any (possibly stochastic) mapping $X \mapsto Z_\Delta$,

$$I(X; Z_\Delta) = H(Z_\Delta) - H(Z_\Delta \mid X) \leq H(Z_\Delta), \tag{18}$$

since $H(Z_\Delta \mid X) \geq 0$, we obtain

$$H(X \mid Z_\Delta) \geq H(X) - H(Z_\Delta). \tag{19}$$

This is the first inequality in Eq. (9). (When the inference-time pipeline is deterministic, $H(Z_\Delta \mid X) = 0$ and (18) holds with equality; we keep the generic inequality to cover potential stochastic components.)

**Step 2: Upper bounding $H(Z_\Delta)$ from differential-entropy bounds.** To connect $H(Z_\Delta)$ with our continuous-entropy analysis, Appendix B models finite precision as a discretizer $Q$ that induces a partition $\{C_k\}$ of $\mathbb{R}^m$. Lemma B.1 provides the exact bridge for any $m$-dimensional continuous $Z$:

$$H\big(Q(Z)\big) = h(Z) + \underbrace{\mathbb{E}\Big[\log \tfrac{1}{\text{vol}\{C_{Q(Z)}\}}\Big] + D[f\|f_Q]}_{\triangleq \kappa_\Delta(Z)}, \tag{20}$$

where $f$ is the pdf of $Z$, $f_Q$ is the histogram density induced by the partition, and $D[f\|f_Q] \geq 0$ is the KL divergence. The correction term $\kappa_\Delta(Z)$ depends on the run-time discretizer (indexed by $\Delta$) and the distribution of $Z$.

To streamline notation in Eq. (9), we introduce *dimension-wise* correction constants $\kappa_\Delta(d)$ and $\kappa_\Delta(D)$ for the $d$- and $D$-dimensional representations considered in this work under the same run-time discretizer $Q$ (i.e., the same precision $\Delta$). Specifically, we choose them such that

$$\kappa_\Delta(d) \geq \kappa_\Delta(Z_{\text{feat}}), \qquad \kappa_\Delta(D) \geq \kappa_\Delta(Z_{\text{dec}}), \tag{21}$$

where $\kappa_\Delta(Z)$ is defined in (20).

**Remark (role of $\kappa_\Delta(\cdot)$).** These constants are introduced solely to simplify the presentation while preserving a valid upper bound under a fixed run-time discretizer. Since they upper-bound $\kappa_\Delta(Z)$ over the encountered representations of a given

dimension, they may be loose; importantly, they act as additive corrections governed by the deployment precision, rather than layer-specific modeling assumptions.

Consequently, for any $m$-dimensional representation $Z$ stored at this precision, we have

$$H(Z_\Delta) = H\big(Q(Z)\big) \ \leq \ h(Z) + \kappa_\Delta(m), \tag{22}$$

and in particular this holds with $m = d$ for $Z_{\text{feat}}$ and with $m = D$ for $Z_{\text{dec}}$.

**Practical note.** Because $\kappa_\Delta(m)$ is an upper envelope, the resulting bound can be numerically loose and may become vacuous in absolute value (e.g., after substitution into $H(X \mid Z_\Delta) \geq H(X) - H(Z_\Delta)$). In this work, $\kappa_\Delta(m)$ serves to account for finite-precision effects under a fixed deployment discretizer, while our layer-wise comparisons rely on the variation of the differential-entropy terms rather than tightness of this additive offset.

Under a fixed deployment precision, $\kappa_\Delta(m)$ acts as a precision-dependent additive offset. It does not encode the layerwise geometric contraction captured by $\sigma_{\text{feat}}^2$ and $R_c^2$, so the pronounced increase of the bound across the representational transition is driven primarily by the differential-entropy surrogates rather than by $\kappa_\Delta(m)$.

**Remark: uniform quantization as a special case.** If $Q$ is a coordinate-wise uniform quantizer with step size $\Delta$, then each cell has $\text{vol}\{C_k\} = \Delta^m$, and Eq. (20) specializes to

$$H(Z_\Delta) = H\big(Q(Z)\big) = h(Z) + m \log \tfrac{1}{\Delta} + D[f \| f_\Delta],$$

where $f_\Delta$ denotes the histogram density induced by the uniform partition. This recovers the familiar continuous–discrete entropy bridge. In the high-resolution regime, where the discretization cells are sufficiently fine and the mismatch term $D[f \| f_\Delta]$ is small, the dominant correction is $m \log \tfrac{1}{\Delta}$. Importantly, Eq. (20) is not restricted to uniform quantization and also applies to non-uniform discretizations such as FP32 rounding; hence, Eq. (22) is not tied to the uniform-quantization assumption.

For fixed $\Delta$, the uniform-quantization bridge contains an explicit linear term $m \log \tfrac{1}{\Delta}$ plus a nonnegative mismatch term. Thus, when the high-resolution approximation is reasonable, the leading dependence of the finite-precision correction on dimension is linear in $m$, consistent with treating it as a precision-dependent additive offset.

**Step 3: Instantiating the bound for $Z_{\text{feat}}$ and $Z_{\text{dec}}$.** For the feature-level representation $Z_{\text{feat}} \in \mathbb{R}^d$, Eq. (4) gives

$$h(Z_{\text{feat}}) \leq \frac{d}{2} \ln\big(2\pi e \, \sigma_{\text{feat}}^2\big). \tag{23}$$

Combining (19), (22), and (23) yields

$$H(X \mid Z_{\text{feat},\Delta}) \geq H(X) - \frac{d}{2} \ln\big(2\pi e \, \sigma_{\text{feat}}^2\big) - \kappa_\Delta(d). \tag{24}$$

For the decision-level representation $Z_{\text{dec}} \in \mathbb{R}^D$, Eq. (6) gives

$$h(Z_{\text{dec}}) \leq \frac{D}{2} \ln\Big(2\pi e \, \frac{R_c^2}{D}\Big) + H(Y). \tag{25}$$

Similarly,

$$H(X \mid Z_{\text{dec},\Delta}) \geq H(X) - \frac{D}{2} \ln\Big(2\pi e \, \frac{R_c^2}{D}\Big) - H(Y) - \kappa_\Delta(D). \tag{26}$$

**Step 4: Concluding Eq. (9).** Eqs. (24) and (26) yield the two cases in Eq. (9).

**Ordering of $H(X \mid Z)$ across split depths.** Fix a deployment setting with a fixed run-time precision $\Delta$, and let $Z_{\ell,\Delta}$ denote the run-time activation at layer $\ell$ under this same precision. Consider two split points $\ell_{\text{feat}} < \ell_{\text{dec}}$ and define $Z_{\text{feat},\Delta} := Z_{\ell_{\text{feat}},\Delta}$ and $Z_{\text{dec},\Delta} := Z_{\ell_{\text{dec}},\Delta}$. Under the run-time forward computation, the mapping from layer $\ell_{\text{feat}}$ to layer

*Table 5.* Comparison of representation transition zone location methods.

| Model Architecture | Empirical location | Active location via $R_c^2$ |
|:---:|:---|:---|
| IR-152 | 48→49 | 48→49 |
| IR-152 | 48→49 | (w/o dim. norm.) 48→49 |
| VGG19 | 17→26 and 39→43 | 26→30 and 30→39 |
| VGG19 | 17→26 and 39→43 | (w/o dim. norm.) 26→30 and 39→43 |

$\ell_{\text{dec}}$ is deterministic and uses no additional input beyond $Z_{\text{feat},\Delta}$. Therefore there exists a deterministic function $g$ such that $Z_{\text{dec},\Delta} = g(Z_{\text{feat},\Delta})$, and

$$X \to Z_{\text{feat},\Delta} \to Z_{\text{dec},\Delta}$$

forms a Markov chain. By the data processing inequality,

$$I(X; Z_{\text{feat},\Delta}) \geq I(X; Z_{\text{dec},\Delta}).$$

Using the identity $I(X; Z_\Delta) = H(X) - H(X \mid Z_\Delta)$ and the fact that $H(X)$ is the same for both split points, it follows that

$$H(X \mid Z_{\text{feat},\Delta}) \leq H(X \mid Z_{\text{dec},\Delta}).$$

Consequently, even if the lower bound in Eq. (9) at a shallower split is numerically smaller than that at a deeper split, the true conditional entropies cannot reverse order under this setting, and the inequality is typically strict when the deeper representation discards instance-specific information, which is the regime where the gap becomes pronounced around the transition.

## D. Pseudocode for Active Location of the Golden Partition Zone

The algorithm 1 in Appendix D actively locates the *Golden Partition Zone* by tracking how the intra-class mean-squared radius $R_c^{2(l)}$ evolves across layers. Our motivation is dynamics-driven: once the intermediate representation $z^{(l)}$ becomes decision-level, contracting the intra-class radius is aligned with the late-stage training objective. This mechanism is supported by Appendix E and F. Consequently, compared with feature-level representations, decision-level representations tend to exhibit a more pronounced decrease in $R_c^{2(l)}$, producing a stable peak signal for locating the transition.

The input is a mini-batch of samples that may contain one or multiple classes. For a single-class batch, $R_c^{2(l)}$ is computed directly. For a multi-class batch, we compute $R_c^{2(l)}$ per class and average across classes; empirically, this averaging preserves the transition peak and improves robustness. To avoid spurious effects caused by changing representation dimensionality across layers, we further normalize the radius by the layer dimension $d_l$, and perform all drop calculations using the normalized radius $\widetilde{R}_c^{2(l)} \triangleq R_c^{2(l)}/d_l$.

To locate the transition zone, the algorithm computes the percentage decrease of the (dimension-normalized) intra-class radius between adjacent target layers. Specifically, for a layer $l$ with representation dimension $d_l$, we use the normalized radius $\widetilde{R}_c^{2(l)} \triangleq R_c^{2(l)}/d_l$ and define

$$R_{\text{drop}}^{2(l)} = \frac{\widetilde{R}_c^{2(l_{\text{previous}})} - \widetilde{R}_c^{2(l_{\text{current}})}}{\widetilde{R}_c^{2(l_{\text{previous}})}} \times 100\%. \tag{27}$$

The layer attaining the maximum $R_{\text{drop}}^{2(l)}$ is selected as the transition-peak layer $l_{\text{TP}}$. The transition-start layer $l_{\text{TS}}$ is chosen as the largest drop preceding $l_{\text{TP}}$ in the target-layer list, so that $l_{\text{TS}}$ is temporally consistent with the peak.

Finally, the algorithm inspects the intermediate layers between $l_{\text{TS}}$ and $l_{\text{TP}}$. If the drop magnitudes of all intermediate layers stay below a threshold (e.g., 20%), the transition is treated as localized at $l_{\text{TP}}$; otherwise, the transition zone is defined as the contiguous span from $l_{\text{TS}}$ to $l_{\text{TP}}$.

The feasibility of the proposed algorithm is further validated experimentally. Table 5 compares transition zones identified by empirical inspection with those obtained automatically via $R_c^2$. The results show that the active procedure is consistent

with empirical observations on IR-152, and for VGG19 it yields a more focused subset within the empirically verified transition range, providing a systematic and reproducible localization. Importantly, without dimension normalization, the drop score can be biased by changes in representation dimensionality, causing the algorithm to select layers where $d_l$ decreases rather than where the class-conditional geometry truly contracts. After normalizing by $d_l$, the method instead captures the pronounced $R_c^2$ contraction within the constant-dimensional block (e.g., layers 30→39), which better reflects the intended decision-representation transition.

**Stability checks.** We further perform three stability checks to ensure that the automatically located zone is not an artifact of randomness: (i) **Sampling stability.** Under a fixed trained checkpoint, rerunning Algorithm 1 with different evaluation seeds (thus different shuffled mini-batches) yields identical transition indices (agreement 3/3, pairwise Jaccard = 1.00), indicating robustness to batch sampling noise. (ii) **Training-seed stability.** Training the target model in multiple independent runs with different random seeds (including shuffled mini-batch orders) leads to consistent located zones across checkpoints, suggesting robustness to typical training stochasticity. (iii) **Condition stability w.r.t. label smoothing.** The located zone remains unchanged between standard training and label smoothing, indicating that the transition is not driven by this training-condition change.

The implementation of the algorithm is also publicly available at *https://github.com/GoldenPartitionZone/ GoldenPartitionZone*.

# E. First-order change of $R_c^2$ under one-hot and label smoothing

We analyze a virtual gradient-descent step on the representation $z$ to characterize how backpropagated gradients reshape intra-class compactness. This is a standard first-order sensitivity analysis: although SGD updates the network parameters, its effect on the current-layer representations can be locally captured by the backpropagated gradient $\tilde{g}_i := \partial \mathcal{L}/\partial z_i$. As such, the analysis characterizes the instantaneous local drift of $R_c^2$, while the long-term training behavior corresponds to the accumulation of such local effects.

**Setup and notation.** Fix a class $c$ and let $\mathcal{I}_c = \{i : y_i = e_c\}$ be the index set of samples in this class, with $N_c = |\mathcal{I}_c|$. At a given layer $\ell$ (omitted for brevity), each representation satisfies $z_i \in \mathbb{R}^{d_\ell}$, and we define

$$\mu_c := \frac{1}{N_c} \sum_{i \in \mathcal{I}_c} z_i, \qquad R_c^2 := \frac{1}{N_c} \sum_{i \in \mathcal{I}_c} \|z_i - \mu_c\|^2. \tag{28}$$

Consider one gradient step on $z_i$ with step size (learning rate) $\gamma > 0$:

$$z_i^+ = z_i - \gamma\, \tilde{g}_i, \qquad \tilde{g}_i := \frac{\partial \mathcal{L}}{\partial z_i}. \tag{29}$$

The class center updates accordingly:

$$\mu_c^+ = \frac{1}{N_c} \sum_{i \in \mathcal{I}_c} z_i^+ = \mu_c - \gamma\, \tilde{g}_c, \qquad \tilde{g}_c := \frac{1}{N_c} \sum_{i \in \mathcal{I}_c} \tilde{g}_i. \tag{30}$$

**Step 1: expand the updated radius.** By definition,

$$R_c^{2+} := \frac{1}{N_c} \sum_{i \in \mathcal{I}_c} \|z_i^+ - \mu_c^+\|^2. \tag{31}$$

Substituting (29)–(30) gives

$$z_i^+ - \mu_c^+ = (z_i - \mu_c) - \gamma(\tilde{g}_i - \tilde{g}_c).$$

Hence,

$$R_c^{2+} = \frac{1}{N_c} \sum_{i \in \mathcal{I}_c} \left\|(z_i - \mu_c) - \gamma(\tilde{g}_i - \tilde{g}_c)\right\|^2. \tag{32}$$

**Step 2: isolate the first-order term in $\gamma$.** Using $\|a - \gamma b\|^2 = \|a\|^2 - 2\gamma\, a^\top b + \gamma^2\|b\|^2$ with $a = (z_i - \mu_c)$ and $b = (\tilde{g}_i - \tilde{g}_c)$ in (32), we obtain

$$R_c^{2+} = \frac{1}{N_c} \sum_{i \in \mathcal{I}_c} \left( \|z_i - \mu_c\|^2 - 2\gamma\,(z_i - \mu_c)^\top (\tilde{g}_i - \tilde{g}_c) + \gamma^2 \|\tilde{g}_i - \tilde{g}_c\|^2 \right)$$

$$= R_c^2 - \frac{2\gamma}{N_c} \sum_{i \in \mathcal{I}_c} (z_i - \mu_c)^\top (\tilde{g}_i - \tilde{g}_c) + \frac{\gamma^2}{N_c} \sum_{i \in \mathcal{I}_c} \|\tilde{g}_i - \tilde{g}_c\|^2. \tag{33}$$

Now expand the linear term:

$$\sum_{i \in \mathcal{I}_c} (z_i - \mu_c)^\top (\tilde{g}_i - \tilde{g}_c) = \sum_{i \in \mathcal{I}_c} (z_i - \mu_c)^\top \tilde{g}_i - \left( \sum_{i \in \mathcal{I}_c} (z_i - \mu_c) \right)^\top \tilde{g}_c$$

$$= \sum_{i \in \mathcal{I}_c} (z_i - \mu_c)^\top \tilde{g}_i, \tag{34}$$

where we used the identity $\sum_{i \in \mathcal{I}_c} (z_i - \mu_c) = 0$. Combining (33)–(34), the radius increment satisfies

$$\Delta R_c^2 := R_c^{2+} - R_c^2 = -\frac{2\gamma}{N_c} \sum_{i \in \mathcal{I}_c} (z_i - \mu_c)^\top \tilde{g}_i \; + \; O(\gamma^2). \tag{35}$$

We denote the first-order term by

$$\Delta R_{c,\mathrm{1st}}^2 := -\frac{2\gamma}{N_c} \sum_{i \in \mathcal{I}_c} (z_i - \mu_c)^\top \tilde{g}_i, \qquad \Delta R_c^2 = \Delta R_{c,\mathrm{1st}}^2 + O(\gamma^2). \tag{36}$$

**Step 3: express $\tilde{g}_i$ via logits and Jacobian.** Let $o_i \in \mathbb{R}^K$ be the logits and $p_i = \mathrm{softmax}(o_i)$. Define the Jacobian $J_i := \partial o_i / \partial z_i \in \mathbb{R}^{K \times d_\ell}$. For any loss whose logit-gradient is $\delta_i := \partial \mathcal{L} / \partial o_i \in \mathbb{R}^K$, the chain rule gives

$$\tilde{g}_i = \frac{\partial \mathcal{L}}{\partial z_i} = J_i^\top \delta_i. \tag{37}$$

We further define the projection terms

$$T_{\mathrm{corr},i} := (z_i - \mu_c)^\top J_i^\top e_c, \qquad T_{k,i} := (z_i - \mu_c)^\top J_i^\top e_k \ (k \neq c), \tag{38}$$

where $\{e_k\}_{k=1}^K$ are standard basis vectors in $\mathbb{R}^K$.

### E.1. One-hot supervision

For a sample in class $c$, the cross-entropy logit-gradient is

$$\delta_i = p_i - y_i = p_i - e_c, \quad \text{so that} \quad \delta_{ic} = p_{ic} - 1, \; \delta_{ik} = p_{ik} \ (k \neq c).$$

Therefore, (37) becomes

$$\tilde{g}_i = J_i^\top (p_i - e_c) = (p_{ic} - 1) J_i^\top e_c + \sum_{k \neq c} p_{ik} J_i^\top e_k. \tag{39}$$

Substituting (39) into (36) and using (38) yields

$$\Delta R_{c,\mathrm{1st}}^2 (\text{one-hot}) = -\frac{2\gamma}{N_c} \sum_{i \in \mathcal{I}_c} (z_i - \mu_c)^\top \left[ (p_{ic} - 1) J_i^\top e_c + \sum_{k \neq c} p_{ik} J_i^\top e_k \right]$$

$$= -\frac{2\gamma}{N_c} \sum_{i \in \mathcal{I}_c} \left[ (p_{ic} - 1) T_{\mathrm{corr},i} + \sum_{k \neq c} p_{ik} T_{k,i} \right]. \tag{40}$$

This is the first-order expression reported in the main text (up to $O(\gamma^2)$).

## E.2. Label smoothing

Under label smoothing, for a sample in class $c$ the smoothed target is

$$y'_{ic} = 1 - \alpha, \qquad y'_{ik} = \frac{\alpha}{K-1} \ (k \neq c),$$

and the logit-gradient becomes

$$\tilde{\delta}_i = p_i - y'_i, \quad \text{so that} \quad \tilde{\delta}_{ic} = p_{ic} - 1 + \alpha, \ \tilde{\delta}_{ik} = p_{ik} - \frac{\alpha}{K-1} \ (k \neq c).$$

Hence,

$$\tilde{g}_i^{\text{LS}} = J_i^\top \tilde{\delta}_i = (p_{ic} - 1 + \alpha) J_i^\top e_c + \sum_{k \neq c} \left( p_{ik} - \frac{\alpha}{K-1} \right) J_i^\top e_k. \tag{41}$$

Substituting (41) into (36) gives

$$\Delta R_{c,\text{1st}}^2(\text{LS}) = -\frac{2\gamma}{N_c} \sum_{i \in \mathcal{I}_c} (z_i - \mu_c)^\top \left[ (p_{ic} - 1 + \alpha) J_i^\top e_c + \sum_{k \neq c} \left( p_{ik} - \frac{\alpha}{K-1} \right) J_i^\top e_k \right]$$

$$= -\frac{2\gamma}{N_c} \sum_{i \in \mathcal{I}_c} \left[ (p_{ic} - 1 + \alpha) T_{\text{corr},i} + \sum_{k \neq c} \left( p_{ik} - \frac{\alpha}{K-1} \right) T_{k,i} \right], \tag{42}$$

which recovers Eq. (12) in the main text under label smoothing, up to $O(\gamma^2)$.

**Remark (first-order nature).** Equations (40) and (42) characterize the first-order change of $R_c^2$ induced by backpropagated gradients. All omitted terms are of order $O(\gamma^2)$ and vanish for sufficiently small step size.

## E.3. Prior-weighted smoothing (Szegedy et al., 2016) as a variant of label smoothing

Let $\pi \in \mathbb{R}^K$ denote the class prior of the training data, with $\pi_k \geq 0$ and $\sum_{k=1}^K \pi_k = 1$. For a sample in class $c$, define the normalized off-class prior

$$\rho_k^{(c)} := \frac{\pi_k}{1 - \pi_c} \ (k \neq c), \qquad \rho_c^{(c)} := 0,$$

so that $\sum_{k \neq c} \rho_k^{(c)} = 1$. Class-prior smoothing replaces the uniform off-class mass in label smoothing by this prior-weighted distribution. The smoothed target is

$$y_{ic}^\pi = 1 - \alpha, \qquad y_{ik}^\pi = \alpha \rho_k^{(c)} \ (k \neq c).$$

The logit-gradient becomes

$$\delta_i^\pi = p_i - y_i^\pi, \quad \text{so that} \quad \delta_{ic}^\pi = p_{ic} - 1 + \alpha, \ \delta_{ik}^\pi = p_{ik} - \alpha \rho_k^{(c)} \ (k \neq c).$$

Hence,

$$\tilde{g}_i^\pi = J_i^\top \delta_i^\pi = (p_{ic} - 1 + \alpha) J_i^\top e_c + \sum_{k \neq c} \left( p_{ik} - \alpha \rho_k^{(c)} \right) J_i^\top e_k. \tag{43}$$

Substituting (43) into (36) and using (38) yields

$$\Delta R_{c,\text{1st}}^2(\text{prior}) = -\frac{2\gamma}{N_c} \sum_{i \in \mathcal{I}_c} \left[ (p_{ic} - 1 + \alpha) T_{\text{corr},i} + \sum_{k \neq c} \left( p_{ik} - \alpha \rho_k^{(c)} \right) T_{k,i} \right]. \tag{44}$$

Uniform label smoothing is recovered as a special case when $\pi$ is uniform, since then $\rho_k^{(c)} = 1/(K-1)$ for all $k \neq c$.

**Remark: an $O(\alpha)$ prediction-independent residual under target-distribution reshaping.** Appendix E.2 and E.3 are instances of a broader family of target reshaping schemes that replace the one-hot target $y_i = e_c$ by a softened target $q^{(c)}$ satisfying $q_c^{(c)} = 1 - \alpha$ and $\sum_{k \neq c} q_k^{(c)} = \alpha$. Equivalently, $q^{(c)}$ can be written as

$$q^{(c)} = (1 - \alpha)e_c + \alpha\, r^{(c)}, \qquad r_c^{(c)} = 0, \quad \sum_{k \neq c} r_k^{(c)} = 1, \tag{45}$$

where $r^{(c)}$ specifies how the off-class mass is distributed and is independent of the model prediction $p_i$. The logit residual then admits the exact decomposition

$$\delta_i^{\text{soft}} := p_i - q^{(c)} = (p_i - e_c) + (e_c - q^{(c)}) = \delta_i^{\text{onehot}} + b^{(c)}, \qquad b^{(c)} := e_c - q^{(c)} = \alpha\,(e_c - r^{(c)}), \tag{46}$$

where $b^{(c)}$ is an $O(\alpha)$ vector that does not depend on $p_i$. Consequently, in the high-confidence regime where $p_i$ is close to $e_c$, the one-hot residual $\delta_i^{\text{onehot}}$ becomes small, while $\delta_i^{\text{soft}}$ approaches the non-vanishing offset $b^{(c)}$.

Substituting (46) into (37) and (36) yields

$$\Delta R_{c,\text{1st}}^2(\text{soft}) = \Delta R_{c,\text{1st}}^2(\text{onehot}) - \frac{2\gamma}{N_c} \sum_{i \in \mathcal{I}_c} (z_i - \mu_c)^\top J_i^\top b^{(c)}$$

$$= \Delta R_{c,\text{1st}}^2(\text{onehot}) - \frac{2\gamma\alpha}{N_c} \sum_{i \in \mathcal{I}_c} \left[ T_{\text{corr},i} - \sum_{k \neq c} r_k^{(c)} T_{k,i} \right]. \tag{47}$$

Thus, any prediction-independent target reshaping of the form (45) adds an extra first-order drive term of order $O(\alpha)$ to the $R_c^2$ dynamics through the same projection terms $T_{\text{corr},i}$ and $T_{k,i}$. This explains why such schemes can continue to influence the $R_c^2$ dynamics even when one-hot training would yield a near-zero logit residual.

### E.4. Angle interpretation of $T_{\text{corr},i}$

We present a geometric interpretation that links the sign of $T_{\text{corr},i}$ to a prevailing phenomenon in supervised classification: within each class, representations often drift inward toward the class center as training progresses. Intuitively, because all samples in class $c$ are trained against the same target distribution $q^{(c)}$, their correct-class residuals share a common structure, making the induced representation updates more coherent within the class; this coherence encourages feature homogenization and an inward drift toward $\mu_c$.

**Geometry of $T_{\text{corr},i}$.** Let $r_i := z_i - \mu_c$ denote the outward direction from the class center. Define $\theta_{c,i}$ as the angle between the correct-logit increasing direction $J_i^\top e_c$ and $r_i$. Then

$$T_{\text{corr},i} = r_i^\top J_i^\top e_c = \|r_i\|\, \|J_i^\top e_c\|\, \cos\theta_{c,i}.$$

Thus, $\theta_{c,i} > 90°$ is equivalent to $T_{\text{corr},i} < 0$: increasing the correct logit has a negative radial projection and therefore points partly inward, toward $\mu_c$.

**Correct-class update and its radial projection.** Recall the virtual representation update $z_i^+ - z_i = -\gamma \tilde{g}_i$. Under label smoothing with $q_c^{(c)} = 1 - \alpha$, the correct-class component is

$$-\gamma\,(p_{ic} - q_c^{(c)})\, J_i^\top e_c = -\gamma\,(p_{ic} - 1 + \alpha)\, J_i^\top e_c.$$

Projecting this component onto $r_i$ yields the radial-change identity

$$r_i^\top (z_i^+ - z_i) = -\gamma\,(p_{ic} - 1 + \alpha)\, T_{\text{corr},i}. \tag{48}$$

The prevailing inward tendency corresponds to requiring $r_i^\top (z_i^+ - z_i) < 0$.

**Under-confidence: $p_{ic} < 1 - \alpha$.** In the one-hot case, $q_c^{(c)} = 1$ so $p_{ic} - q_c^{(c)} = p_{ic} - 1 < 0$ always. Under label smoothing, the same sign holds in the under-confident regime $p_{ic} < 1 - \alpha$, where $p_{ic} - 1 + \alpha < 0$. In both situations, the correct-class term moves along $+J_i^\top e_c$, increasing the correct logit. Consistency with inward drift requires $r_i^\top (z_i^+ - z_i) < 0$; by (48) and $p_{ic} - 1 + \alpha < 0$, this implies $T_{\text{corr},i} < 0$, equivalently $\theta_{c,i} > 90°$. Geometrically, when the model is under-confident, increasing the correct logit is compatible with an inward move only if the correct-logit direction points against the outward radial direction.

**Over-confidence:** $p_{ic} > 1 - \alpha$**.** In the over-confident regime, $p_{ic} - 1 + \alpha > 0$ and the correct-class term can be rewritten as

$$-\gamma \left(p_{ic} - 1 + \alpha\right) J_i^\top e_c = \gamma \left(p_{ic} - 1 + \alpha\right) \left(-J_i^\top e_c\right),$$

so the update moves along $-J_i^\top e_c$, decreasing the correct logit and calibrating over-confidence. Requiring the same inward drift $r_i^\top (z_i^+ - z_i) < 0$ and using (48) with $p_{ic} - 1 + \alpha > 0$ now yields $T_{\mathrm{corr},i} > 0$, equivalently $\theta_{c,i} < 90°$. Therefore the threshold $p_{ic} = 1 - \alpha$ flips which angle alignment is compatible with inward motion: under-confidence prefers $\theta_{c,i} > 90°$, whereas over-confidence prefers $\theta_{c,i} < 90°$. Importantly, this flip does not imply an expansion of $R_c^2$; it only changes whether inward motion is achieved by moving along $+J_i^\top e_c$ or along $-J_i^\top e_c$.

**High-confidence limit.** In the high-confidence limit $p_i \to e_c$, (47) shows that soft targets retain a non-vanishing first-order drive:

$$\Delta R_{c,\mathrm{1st}}^2(\mathrm{soft}) \approx -\frac{2\gamma\alpha}{N_c} \sum_{i \in \mathcal{I}_c} \left[ T_{\mathrm{corr},i} - \sum_{k \neq c} r_k^{(c)} T_{k,i} \right].$$

In this regime, $p_{ic} > 1 - \alpha$ typically holds, so inward drift is naturally compatible with $\theta_{c,i} < 90°$ (i.e., $T_{\mathrm{corr},i} > 0$). Moreover, the weighted off-class term is a convex average over many directions and its radial projections tend to cancel out, so it rarely dominates the bracketed quantity. Consequently, the bracket typically remains positive, the first-order drive stays negative, and $R_c^2$ can keep decreasing even when $p_{ic}$ is close to 1.

**Remark ($\alpha < 0$).** When $\alpha < 0$, the soft target satisfies $q_c^{(c)} = 1 - \alpha > 1$ and $q_k^{(c)} = \alpha/(K - 1) < 0$ for $k \neq c$, so the regime $p_{ic} > 1 - \alpha$ cannot occur since $p_{ic} \leq 1$. Equivalently, the correct-class residual $\tilde{\delta}_{ic} = p_{ic} - 1 + \alpha$ is always negative. Under inward drift, (48) then requires $T_{\mathrm{corr},i} < 0$ (i.e., $\theta_{c,i} > 90°$), so the sign flip discussed above disappears.

## F. Gradient-norm lower bound under label smoothing

This appendix formalizes why label smoothing can prevent the logit-level residuals from vanishing prematurely along over-confident trajectories, and consequently keeps the feature-side gradients from collapsing too early in the late stage of training. Throughout, we use the standard LS targets: for a sample with true class $c$, $y_c' = 1 - \alpha$ and $y_k' = \alpha/(K - 1)$ for all $k \neq c$.

**Setup.** Let $o \in \mathbb{R}^K$ be the logit vector at the cloud side and $p = \mathrm{softmax}(o)$ the predicted probabilities. Let $z_\ell \in \mathbb{R}^{d_\ell}$ denote the representation at layer $\ell$ and define the Jacobian

$$J_\ell := \frac{\partial o}{\partial z_\ell} \in \mathbb{R}^{K \times d_\ell}.$$

For cross-entropy loss, the residual at the logit level equals

$$\delta := \frac{\partial \mathcal{L}_{\mathrm{CE}}}{\partial o} = p - y, \qquad \tilde{\delta} := \frac{\partial \mathcal{L}_{\mathrm{LS}}}{\partial o} = p - y'.$$

The corresponding gradients with respect to the feature representation are

$$g_\ell := \frac{\partial \mathcal{L}_{\mathrm{CE}}}{\partial z_\ell} = J_\ell^\top \delta, \qquad \tilde{g}_\ell := \frac{\partial \mathcal{L}_{\mathrm{LS}}}{\partial z_\ell} = J_\ell^\top \tilde{\delta}.$$

**Step 1: a singular-value lower bound.** Let the SVD of $J_\ell$ be $J_\ell = U\Sigma V^\top$, where $\Sigma = \mathrm{diag}(\sigma_1, \ldots, \sigma_r)$ with $\sigma_1 \geq \cdots \geq \sigma_r > 0$ and $r = \mathrm{rank}(J_\ell)$. For any vector $v \in \mathbb{R}^K$ that lies in the column space of $J_\ell$, one has

$$\|J_\ell^\top v\|_2 = \|\Sigma U^\top v\|_2 \geq \sigma_r \|v\|_2. \tag{49}$$

More generally, for an arbitrary $v \in \mathbb{R}^K$,

$$\|J_\ell^\top v\|_2 \geq \sigma_r \|\Pi_{\mathrm{col}(J_\ell)} v\|_2, \tag{50}$$

where $\Pi_{\mathrm{col}(J_\ell)}$ is the orthogonal projection onto $\mathrm{col}(J_\ell)$. In particular, if $\sigma_r$ does not collapse to zero, then a non-vanishing component of $v$ inside $\mathrm{col}(J_\ell)$ implies a non-vanishing lower bound on $\|J_\ell^\top v\|_2$.

**Step 2: residual norms on an over-confident trajectory.** For a sample whose true class is $c$, define the total off-class mass

$$\varepsilon := 1 - p_c = \sum_{k \neq c} p_k \in [0, 1].$$

Under label smoothing, $\tilde{\delta} = p - y'$ satisfies

$$\|\tilde{\delta}\|_2^2 = (p_c - 1 + \alpha)^2 + \sum_{k \neq c}\left(p_k - \tfrac{\alpha}{K-1}\right)^2 = (\alpha - \varepsilon)^2 + \sum_{k \neq c}\left(p_k - \tfrac{\alpha}{K-1}\right)^2.$$

Using Cauchy–Schwarz,

$$\sum_{k \neq c}\left(p_k - \tfrac{\alpha}{K-1}\right)^2 \geq \frac{1}{K-1}\left(\sum_{k \neq c} p_k - \alpha\right)^2 = \frac{(\varepsilon - \alpha)^2}{K-1}.$$

Therefore,

$$\|\tilde{\delta}\|_2^2 \geq (\alpha - \varepsilon)^2\left(1 + \frac{1}{K-1}\right) = (\alpha - \varepsilon)^2 \frac{K}{K-1}, \qquad \|\tilde{\delta}\|_2 \geq |\alpha - \varepsilon|\sqrt{\frac{K}{K-1}}. \tag{51}$$

*Interpretation.* When the model is over-confident relative to the LS target, namely $\varepsilon < \alpha$, Eq. (51) gives a strictly positive lower bound $\|\tilde{\delta}\|_2 \geq (\alpha - \varepsilon)\sqrt{K/(K-1)}$. Along the limiting one-hot trajectory where $\varepsilon \to 0$, this yields $\|\tilde{\delta}\|_2 \geq c_\alpha := \alpha\sqrt{K/(K-1)}$. At the LS optimum where $p \to y'$, one has $\varepsilon \to \alpha$ and thus $\tilde{\delta} \to 0$ as expected.

For completeness, under one-hot supervision one has $\delta = p - e_c$, hence

$$\|\delta\|_2^2 = (p_c - 1)^2 + \sum_{k \neq c} p_k^2 = \varepsilon^2 + \sum_{k \neq c} p_k^2 \leq \varepsilon^2 + \left(\sum_{k \neq c} p_k\right)^2 = 2\varepsilon^2, \qquad \|\delta\|_2 \leq \sqrt{2}\,\varepsilon. \tag{52}$$

Thus, as the model becomes increasingly over-confident in the one-hot sense and $\varepsilon \to 0$, the logit residual $\delta$ necessarily vanishes.

**Step 3: feature-gradient norms in the over-confident regime.** Applying (50) with $v = \delta$ and with $v = \tilde{\delta}$ yields the generic lower bounds

$$\|g_\ell\|_2 = \|J_\ell^\top \delta\|_2 \geq \sigma_r \|\Pi_{\mathrm{col}(J_\ell)}\delta\|_2, \qquad \|\tilde{g}_\ell\|_2 = \|J_\ell^\top \tilde{\delta}\|_2 \geq \sigma_r \|\Pi_{\mathrm{col}(J_\ell)}\tilde{\delta}\|_2,$$

where $\sigma_r > 0$ is the smallest non-zero singular value of $J_\ell$ and $\Pi_{\mathrm{col}(J_\ell)}$ is the orthogonal projection onto $\mathrm{col}(J_\ell)$.

To make the lower bounds interpretable, define the projection retention factors

$$\kappa_{\ell,i}^{\mathrm{CE}} := \frac{\|\Pi_{\mathrm{col}(J_\ell)}\delta\|_2}{\|\delta\|_2} \in [0, 1], \qquad \kappa_{\ell,i}^{\mathrm{LS}} := \frac{\|\Pi_{\mathrm{col}(J_\ell)}\tilde{\delta}\|_2}{\|\tilde{\delta}\|_2} \in [0, 1],$$

with the convention that the ratio is set to 0 when the denominator is 0. Let $\Pi := \Pi_{\mathrm{col}(J_\ell)}$. Since $\Pi$ is an orthogonal projection, the decompositions $\delta = \Pi\delta + (I - \Pi)\delta$ and $\tilde{\delta} = \Pi\tilde{\delta} + (I - \Pi)\tilde{\delta}$ are Pythagorean, hence $\|\Pi\delta\|_2 \leq \|\delta\|_2$ and $\|\Pi\tilde{\delta}\|_2 \leq \|\tilde{\delta}\|_2$, which implies $\kappa_{\ell,i}^{\mathrm{CE}}, \kappa_{\ell,i}^{\mathrm{LS}} \in [0, 1]$.

**Lower bound in the over-confident regime.** Combining (50) with (51), for any sample that is over-confident relative to the LS target (i.e., $\varepsilon < \alpha$, equivalently $p_c > 1 - \alpha$), we have

$$\|\tilde{g}_\ell\|_2 \geq \sigma_r \|\Pi\tilde{\delta}\|_2 = \sigma_r \kappa_{\ell,i}^{\mathrm{LS}} \|\tilde{\delta}\|_2 \geq \sigma_r \kappa_{\ell,i}^{\mathrm{LS}} (\alpha - \varepsilon)\sqrt{\frac{K}{K-1}}. \tag{53}$$

Thus, unless $J_\ell$ becomes ill-conditioned (so that $\sigma_r$ collapses) or the projection degenerates (so that $\kappa_{\ell,i}^{\mathrm{LS}} \approx 0$), label smoothing yields an explicit non-vanishing feature-gradient lower bound throughout the over-confident regime $\varepsilon < \alpha$.

**Gradient collapse as $\varepsilon \to 0$.** To show collapse under one-hot supervision, we use an *upper* bound via the operator norm:

$$\|g_\ell\|_2 = \|J_\ell^\top \delta\|_2 \leq \|J_\ell^\top\|_2 \|\delta\|_2 = \|J_\ell\|_2 \|\delta\|_2 \leq \sigma_1(J_\ell)\sqrt{2}\,\varepsilon, \tag{54}$$

where $\|J_\ell\|_2 = \sigma_1(J_\ell)$ is the largest singular value of $J_\ell$ and we used (52). Hence, along an increasingly over-confident one-hot trajectory $\varepsilon \to 0$, the feature-side gradient norm satisfies $\|g_\ell\|_2 \to 0$ provided $\sigma_1(J_\ell)$ remains bounded.

**Full-row-rank simplification.** In the common case where $J_\ell$ has full row rank, $\mathrm{col}(J_\ell) = \mathbb{R}^K$ and hence $\kappa_{\ell,i}^{\mathrm{CE}} = \kappa_{\ell,i}^{\mathrm{LS}} = 1$. Eq. (53) then simplifies to

$$\|\tilde{g}_\ell\|_2 \ \geq \ \sigma_{\min}(J_\ell)\,(\alpha - \varepsilon)\sqrt{\frac{K}{K-1}},$$

where $\sigma_{\min}(J_\ell)$ denotes the smallest (positive) singular value.

**Summary.** One-hot supervision drives the logit residual $\delta$ to zero as $p$ approaches the one-hot target ($\varepsilon \to 0$), and consequently $\|g_\ell\|_2$ vanishes under the mild condition that $\sigma_1(J_\ell)$ does not blow up. Label smoothing does not enforce a non-zero gradient at the LS optimum where $p = y'$ (so $\tilde{\delta} = 0$), but it yields an explicit lower bound on $\|\tilde{g}_\ell\|_2$ throughout the over-confident regime $p_c > 1 - \alpha$ (equivalently $\varepsilon < \alpha$) via (51)–(53). Therefore, LS can keep gradients active during late-stage over-confident training unless $J_\ell$ becomes ill-conditioned or the projection term degenerates.

# G. Intuitive interpretation for the *Neural Vortex* metaphor

This section provides only an intuitive interpretation of the term *Neural Vortex*. The discussion is metaphorical and serves to illustrate the training dynamics analyzed in the main text and Appendices E–F. It should not be read as a standalone formal mechanism, a physical model of learning, or an independent source of theoretical or empirical evidence.

**Metaphor.** Consider a liquid mixture whose global disorder increases after stirring, which corresponds to a higher entropy state. Stirring can also induce a coherent local flow structure that is commonly described as a vortex. Although the mixture becomes globally more mixed, the vortex region can exhibit stronger local organization than its surroundings. We use this contrast between global entropy increase and local structural organization to motivate the name *Neural Vortex*.

**Mapping to our setting.** In our setting, reshaping the label distribution, for example by label smoothing, increases output-side uncertainty and discourages over-confident predictions. This corresponds to increasing the entropy of the predictive distribution. At the same time, the decision-level representations can become more compact, which is reflected by a smaller $R_c^2$. Here, entropy refers to uncertainty in the output distribution, while local organization refers to geometric tightness in representation space. This coupling between higher output-side entropy and tighter decision-level geometry is the intuition behind the term *Neural Vortex*.

# H. Supplement for Experiment

## H.1. Datasets

We use seven image datasets spanning natural objects, human faces, and handwritten symbols. All samples are preprocessed to a unified spatial resolution of $64 \times 64$ to standardize the attack and evaluation pipeline across data types. The data allocation is reported in Table 2.

- **CIFAR-10 (Krizhevsky et al., 2009).** CIFAR-10 contains 60,000 natural RGB images from 10 classes. We used the public $64 \times 64$ release[3]. Moreover, the CAI-super-resolution procedure is not simple pixel duplication or naive interpolation.

- **FaceScrub (Ng & Winkler, 2014).** FaceScrub is a web-collected face dataset with on the order of $10^5$ RGB images from 530 identities. However, since not every URL was available during our data acquisition period, we downloaded a total of 43,149 images spanning all 530 individuals. We follow the cropping protocol in (Yang et al., 2019) and resize all images to $64 \times 64$.

- **CelebA (Liu et al., 2015).** CelebA contains 202,599 RGB face images spanning 10,177 identities. We adopt the same preprocessing protocol as (Yang et al., 2019) (cropping and resizing to $64 \times 64$), and ensure there is no identity overlap between FaceScrub and CelebA in our setup.

---

[3] https://www.kaggle.com/datasets/joaopauloschuler/cifar10-64x64-resized-via-cai-super-resolution

*Table 6.* Supplementary target-model configuration and intermediate feature dimensions for ResNet-50 on CIFAR-10.

| Model allocation and accuracy | | |
| --- | --- | --- |
| Task | Model | Accuracy |
| CIFAR-10 | ResNet-50 | One-hot: 86.11% |
| **Dimensions of information at different model depths** | | |
| Model | Block | Dimensions |
| ResNet-50 | $19, 20;\ 21;\ 37, 38;\ 39;\ 46, 47;\ 48$ | $(128, 8, 8);\ (512, 8, 8);\ (256, 4, 4);\ (1024, 4, 4);\ (512, 2, 2);\ (2048, 2, 2)$ |

- **MNIST (LeCun, 1998).** MNIST is a grayscale handwritten digit dataset with 10 classes (0–9).

- **EMNIST (Letters) (Cohen et al., 2017).** EMNIST extends MNIST-style data to handwritten characters. We use the Letters split with 26 classes.

- **KMNIST (Kuzushiji-49) (Clanuwat et al., 2018).** We use the Kuzushiji-49 dataset, which contains 49 grayscale classes (48 Kuzushiji characters plus one iteration mark). For notational convenience, we refer to Kuzushiji-49 as KMNIST throughout the paper.

## H.2. Target Models

In addition to the target models described in the main text, we include **ResNet-50** as an extra target model. ResNet-50 is built from bottleneck residual blocks with an explicit channel expansion factor (`expansion = 4`). Concretely, each block first projects the input to a lower-dimensional feature space via a $1 \times 1$ convolution (from `in_channels` to `out_channels`), processes features with a $3 \times 3$ convolution, and then expands the representation with a final $1 \times 1$ convolution to $4 \times$ `out_channels` channels. Therefore, the post-addition activation (i.e., the output after the residual addition and ReLU) has a larger channel dimension than the intermediate bottleneck features. When the expanded output dimension differs from the identity branch, a downsampling/projection layer is applied to the skip path to match dimensions before the residual addition. This within-block dimension expansion provides a structured change in intermediate representation dimensionality.

## H.3. Inversion Models

Our inversion architecture follows the design principles of (Yang et al., 2019) and (Zhang et al., 2023), and we adaptively strengthen the inversion capacity according to the depth and dimensionality of the target representation. In particular, when the spatial resolution drops to $4 \times 4$, the combination of deeper inversion stacks and `Tanh` activation can lead to vanishing gradients; to mitigate this issue, we replace `Tanh` with `PReLU` (He et al., 2015). Additionally, we incorporate (i) decoupled enhancement strategies and (ii) an inversion-specific architectural redesign. The corresponding analysis and ablations are provided in Section 4.2 (Part 3). The full inversion architecture is available at GitHub.

## H.4. Implementation Details

Target models are trained with Adam (Kingma & Ba, 2014) using a learning rate of $3 \times 10^{-4}$, except for VGG19 on FaceScrub, which is trained with SGD using a learning rate of $0.05$. Inversion models are trained with Adam using $\beta_1 = 0.5$ and a learning rate of $2 \times 10^{-4}$. All training employs `ReduceLROnPlateau`. Experiments are run on two NVIDIA RTX 4090 GPUs and an Intel Core i9-14900KF CPU (32 threads).

**$H(Z)$ enhancement modules.** To assess whether strengthening intermediate information can improve inversion and to stress-test the robustness of the decision-level zone, we implement three lightweight feature enhancement modules (Table 3). Given an intermediate representation $z \in \mathbb{R}^{B \times C \times H \times W}$, we construct an enhanced feature $\tilde{z}$ and combine it with $z$ either by residual addition ($\tilde{z} \leftarrow z + \Delta(z)$) or by channel-wise concatenation ($\tilde{z} \leftarrow \mathrm{Concat}(z, \Delta(z))$), depending on the ablation setting.

**(i) FFT-based enhancement.** We compute a 2D orthonormal FFT over spatial dimensions and extract a stabilized spectral-energy feature:

$$\Delta_{\mathrm{FFT}}(z) = \mathrm{Norm}\Big( \log\big(1 + |\Re(\mathcal{F}(z))| + |\Im(\mathcal{F}(z))|\big)\Big),$$

Here $\mathcal{F}(\cdot)$ denotes the 2D FFT that maps a real feature map to a complex spectrum. We use $\Re(\mathcal{F}(z))$ and $\Im(\mathcal{F}(z))$ to denote its real and imaginary components, respectively, which correspond to `fft.real` and `fft.imag` in our implementation. Concretely, for each channel $c$ and spatial frequency index $(u, v)$, the orthonormal 2D FFT coefficient is

$$\mathcal{F}(z)_{c,u,v} = \frac{1}{\sqrt{HW}} \sum_{h=0}^{H-1} \sum_{w=0}^{W-1} z_{c,h,w} \exp\left(-j\, 2\pi\left(\tfrac{uh}{H} + \tfrac{vw}{W}\right)\right),$$

so that $\mathcal{F}(z)_{c,u,v} = A_{c,u,v} + jB_{c,u,v}$ with

$$\Re(\mathcal{F}(z))_{c,u,v} = A_{c,u,v}, \qquad \Im(\mathcal{F}(z))_{c,u,v} = B_{c,u,v}.$$

Equivalently, letting $\theta_{u,v}(h, w) = 2\pi(\tfrac{uh}{H} + \tfrac{vw}{W})$, we have

$$\Re(\mathcal{F}(z))_{c,u,v} = \sum_{h,w} z_{c,h,w} \cos\bigl(\theta_{u,v}(h, w)\bigr), \quad \Im(\mathcal{F}(z))_{c,u,v} = -\sum_{h,w} z_{c,h,w} \sin\bigl(\theta_{u,v}(h, w)\bigr).$$

To avoid scale explosion and distribution shift, we apply per-sample standardization $\mathrm{Norm}(u) = (u - \mu(u))/(\sigma(u) + 10^{-6})$, where $\mu(\cdot)$ and $\sigma(\cdot)$ are computed over $(C, H, W)$. Unless otherwise stated, we use residual addition $\tilde{z} = z + \Delta_{\mathrm{FFT}}(z)$.

**(ii) Global normalization enhancement.** We implement a feature normalizer that standardizes $z$ using pre-computed channel-wise mean and standard deviation:

$$\Delta_{\mathrm{Norm}}(z) = \frac{z - \mu}{\sigma}, \qquad \mu, \sigma \in \mathbb{R}^C,$$

where $\mu$ and $\sigma$ are stored as buffers and broadcast to $(1, C, 1, 1)$. This module is used either as an input-level normalization prior to inversion, or as an auxiliary branch that is combined with $z$ via residual addition or concatenation.

**(iii) NN-based enhancement.** We implement a compact autoencoder-style enhancement network that compresses and then reconstructs features: two strided $3 \times 3$ convolutions reduce the channel width ($C \rightarrow C/2 \rightarrow C/4$) while downsampling, followed by `BatchNorm+PReLU`, and two transposed convolutions restore the original resolution and channels ($C/4 \rightarrow C/2 \rightarrow C$). The output $\Delta_{\mathrm{NN}}(z)$ is then combined with $z$ by residual addition or concatenation. When reported, dropout is applied inside this enhancement branch to mitigate overfitting of the inversion model.

### H.5. Supplementary Analysis for Residual Networks and VGG

**(i) Analysis for Residual Networks.** Table 7 reports the exact numerical results underlying Fig. 4 in the main text, and further supplements the reconstruction performance on the auxiliary set.

For Blocks 25, 40, and 48, the feature dimensionality remains unchanged at (256, 8, 8) and the representation has not yet undergone a zone shift. Under this fixed-dimensional, pre-transition setting, increasing depth does not yield stronger resistance to MIA. Instead, the inversion performance on both the auxiliary set and the test set can become slightly better at deeper blocks than at Block 25 (highlighted in red), indicating that moving the partition deeper is not always a reliable defence.

At Block 48, label smoothing begins to exhibit a noticeable influence: compared with one-hot training, LS already starts to increase the reconstruction error, consistent with our analysis that LS can gradually induce stronger contraction of $R_c^2$ before the transition fully materializes.

After entering the GPZ, the behavior changes qualitatively. Once the spatial resolution drops to $4 \times 4$ (Blocks 49–51), all three blocks share the same dimensionality (512, 4, 4), yet the inversion error increases monotonically with depth on both auxiliary and test sets. This indicates that deeper decision-level representations become progressively harder to invert, and that effective resistance in residual networks emerges only after crossing the GPZ rather than from depth alone.

As shown in Fig. 10, besides IR-152, we also conduct experiments on **ResNet-50**. As introduced in Section H.2 and summarized in Table 6, ResNet-50 exhibits a characteristic $4\times$ output-dimension expansion in its bottleneck blocks. Empirically, when depth increases while the representation dimension remains unchanged (e.g., 19→20, 37→38, and 46→47), MSE increases and PSNR decreases, indicating weaker MIA performance. In contrast, under the same spatial resolution, increasing the channel width (thus enlarging the overall representation dimension) leads to lower MSE and

*Table 7.* Model inversion attack performance on different residual blocks of IR-152 on CIFAR-10. The table reports reconstruction quality on the auxiliary training set and the test set, including MSE, PSNR, SSIM, and LPIPS. Red entries indicate that, under the same feature dimensionality, increasing depth does not degrade reconstruction quality. **Bold** numbers indicate the highest MSE / PSNR within each block across the three training conditions.

| Block (Dim) | Training Condition | Auxiliary Data Set | | | | Test Data Set | | | |
| --- | --- | --- | --- | --- | --- | --- | --- | --- | --- |
| | | MSE | PSNR | SSIM | LPIPS | MSE | PSNR | SSIM | LPIPS |
| 3 (64, 32, 32) | Normal (90.59%) | 0.000951 | 35.039556 | 0.9788 | 0.0072 | **0.000867** | 35.413233 | 0.9799 | 0.0067 |
| | LS ($\alpha = 0.3$) (90.76%) | 0.000868 | **35.440656** | 0.9811 | 0.0067 | 0.000757 | **36.006741** | 0.9824 | 0.0059 |
| | LS ($\alpha = -0.05$) (90.49%) | **0.000966** | 34.971804 | 0.9803 | 0.0066 | 0.000863 | 35.431071 | 0.9812 | 0.0061 |
| 8 (128, 16, 16) | Normal | **0.005040** | 27.783279 | 0.9102 | 0.0468 | **0.004622** | 28.135101 | 0.9126 | 0.0453 |
| | LS ($\alpha = 0.3$) | 0.004081 | **28.699895** | 0.9236 | 0.0387 | 0.003732 | **29.063014** | 0.9253 | 0.0373 |
| | LS ($\alpha = -0.05$) | 0.004898 | 27.907602 | 0.9103 | 0.0458 | 0.004533 | 28.220955 | 0.9112 | 0.0448 |
| 25 (256, 8, 8) | Normal | **0.011994** | 24.009093 | 0.8199 | 0.0992 | **0.018332** | 22.151508 | 0.7845 | 0.1146 |
| | LS ($\alpha = 0.3$) | 0.011375 | **24.238845** | 0.8211 | 0.0961 | 0.017422 | **22.372892** | 0.7901 | 0.1110 |
| | LS ($\alpha = -0.05$) | 0.011810 | 24.074101 | 0.8153 | 0.1004 | 0.018181 | 22.186258 | 0.7828 | 0.1152 |
| 40 (256, 8, 8) | Normal | 0.011796 | 24.080942 | 0.8205 | 0.0966 | 0.018245 | 22.171516 | 0.7848 | 0.1131 |
| | LS ($\alpha = 0.3$) | 0.011313 | **24.260800** | 0.8255 | 0.0918 | 0.017390 | **22.380720** | 0.7913 | 0.1079 |
| | LS ($\alpha = -0.05$) | **0.012229** | 23.926252 | 0.8113 | 0.1010 | **0.018324** | 22.152125 | 0.7798 | 0.1154 |
| 48 (256, 8, 8) | Normal | 0.011440 | 24.213205 | 0.8179 | 0.1000 | **0.018330** | 22.151773 | 0.7832 | 0.1160 |
| | LS ($\alpha = 0.3$) | 0.010778 | **24.470866** | 0.8217 | 0.0935 | 0.017573 | **22.334486** | 0.7906 | 0.1088 |
| | LS ($\alpha = -0.05$) | **0.012018** | 23.997207 | 0.8140 | 0.0988 | 0.018221 | 22.176246 | 0.7822 | 0.1131 |
| 49 (512, 4, 4) | Normal | 0.021301 | **21.514565** | 0.6514 | 0.1985 | 0.055127 | **17.370814** | 0.5327 | 0.2523 |
| | LS ($\alpha = 0.3$) | **0.037924** | 19.005789 | 0.5453 | 0.2872 | **0.066622** | 16.547368 | 0.4511 | 0.3311 |
| | LS ($\alpha = -0.05$) | 0.027733 | 20.372392 | 0.6374 | 0.2071 | 0.058004 | 17.148076 | 0.5146 | 0.2613 |
| 50 (512, 4, 4) | Normal | 0.021366 | **21.500321** | 0.6425 | 0.2069 | 0.057010 | **17.224045** | 0.5212 | 0.2616 |
| | LS ($\alpha = 0.3$) | **0.038196** | 18.972961 | 0.5116 | 0.3236 | **0.076292** | 15.959300 | 0.4139 | 0.3642 |
| | LS ($\alpha = -0.05$) | 0.031728 | 19.785221 | 0.5728 | 0.2631 | 0.060823 | 16.942471 | 0.4971 | 0.2974 |
| 51 (512, 4, 4) | Normal | 0.025284 | 20.769392 | 0.6342 | 0.2124 | 0.056566 | **17.257185** | 0.5192 | 0.2672 |
| | LS ($\alpha = 0.3$) | **0.035250** | 19.323376 | 0.5169 | 0.3217 | **0.078223** | 15.849769 | 0.4089 | 0.3694 |
| | LS ($\alpha = -0.05$) | 0.024915 | **20.832582** | 0.5815 | 0.2552 | 0.061960 | 16.861872 | 0.4892 | 0.2992 |

higher PSNR (e.g., 19→21, 37→39, and 46→48), i.e., stronger inversion attacks. This architecture-specific behavior further demonstrates that the common intuition that "deeper layers inherently impede MIA" is not always correct.

Notably, the relative MSE reductions from 19 to 21, 37 to 39, and 46 to 48 are 38.48%, 29.67%, and 4.81%, respectively. That is, when the $R_c^2$ contraction becomes sufficiently small, the benefit of increasing the representation dimension for MIA diminishes. This trend is consistent with the dependence on $R_c^2/D$ in the lower bound of $H(X \mid Z)$.

**(ii) Analysis for VGG.** Table 8 provides the detailed numerical results corresponding to Fig. 5 in the main text. A notable pattern is that, for multiple pairs of blocks with identical dimensionality—namely 17→26, 30→39, and 43→52—the reconstruction error increases as depth grows (higher MSE and lower PSNR). More importantly, although 17→26 and 30→39 share exactly the same dimensions, the MSE rises substantially, indicating that simply moving to a deeper block can significantly reduce inversion quality even without any explicit dimensional change.

In addition, from Block 30 onward, label smoothing consistently yields higher MSE than one-hot training. This observation aligns with the following mechanism: **one-hot supervision drives the feature-layer gradients toward zero at convergence, effectively freezing shallow representations; in contrast, LS maintains a strictly positive lower bound on those gradients, so these layers continue to receive small but non-zero updates even in the late stage of training (see Appendix F for details). Consequently, early features remain more active and less redundant across channels, which increases their entropy $H(z)$ and makes feature-level representations more susceptible to MIA.** Therefore, the consistently worse inversion performance under LS beyond Block 30 can be interpreted as the stage where LS starts to exert a clear effect on the contraction of $R_c^2$.

Finally, we emphasize that on VGG, inversion can become ineffective even when the probed representations have dimensions comparable to those in IR-152 and ResNet-50. This further indicates that dimensionality alone does not determine inversion difficulty, and that architecture-dependent representational dynamics play a critical role.

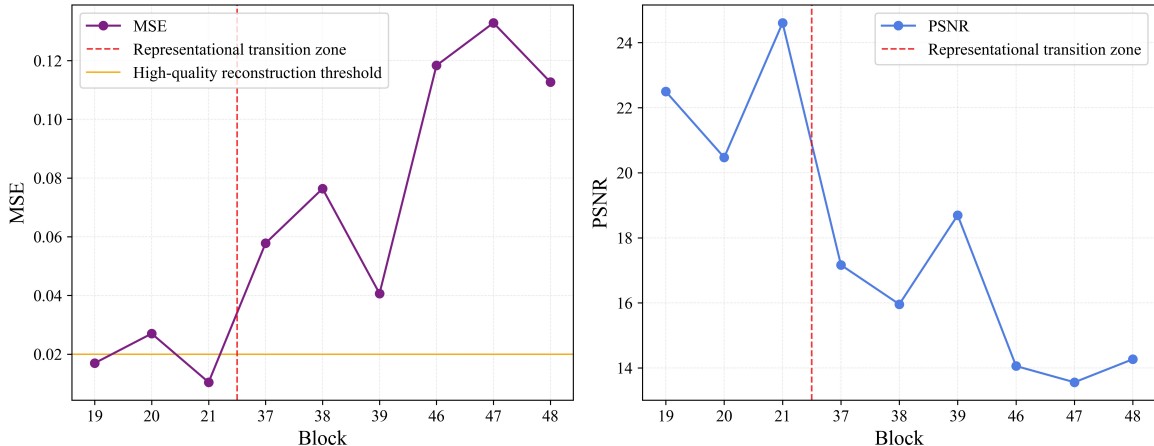

*Figure 10.* MIA performance across different blocks of ResNet-50 on CIFAR-10.

*Table 8.* Model inversion attack performance on different depths of VGG19 on CIFAR-10. The table reports reconstruction quality on the auxiliary training set and the test set, including MSE, PSNR, SSIM, and LPIPS. **Bold** numbers indicate the highest MSE / PSNR within each block across the two training conditions.

| Depth (Dim) | Training Condition | Auxiliary Data Set | | | | Test Data Set | | | |
|---|---|---|---|---|---|---|---|---|---|
| | | MSE | PSNR | SSIM | LPIPS | MSE | PSNR | SSIM | LPIPS |
| 13 (128, 32, 32) | Normal (91.50%) | **0.011416** | 24.229702 | 0.7927 | 0.1083 | **0.011534** | 24.167314 | 0.7905 | 0.1087 |
| | LS ($\alpha = 0.3$) (91.85%) | 0.011030 | **24.387694** | 0.7991 | 0.1051 | 0.011082 | **24.345436** | 0.8004 | 0.1048 |
| 17 (256, 16, 16) | Normal | **0.011754** | 24.101217 | 0.8222 | 0.0965 | **0.013794** | 23.391311 | 0.8091 | 0.1036 |
| | LS ($\alpha = 0.3$) | 0.011007 | **24.390515** | 0.8270 | 0.0954 | 0.013299 | **23.549928** | 0.8128 | 0.1027 |
| 26 (256, 16, 16) | Normal | 0.039417 | **18.838569** | 0.5831 | 0.2598 | **0.049008** | 17.883296 | 0.5612 | 0.2720 |
| | LS ($\alpha = 0.3$) | **0.040755** | 18.693089 | 0.5810 | 0.2559 | 0.048115 | **17.960066** | 0.5529 | 0.2709 |
| 30 (512, 8, 8) | Normal | 0.037669 | **19.039757** | 0.4893 | 0.3477 | 0.066493 | **16.554013** | 0.4455 | 0.3727 |
| | LS ($\alpha = 0.3$) | **0.049440** | 17.855415 | 0.4775 | 0.3562 | **0.067120** | 16.517554 | 0.4309 | 0.3758 |
| 39 (512, 8, 8) | Normal | 0.120302 | **13.997630** | 0.2747 | 0.6320 | 0.136896 | **13.424092** | 0.2533 | 0.6339 |
| | LS ($\alpha = 0.3$) | **0.122318** | 13.929368 | 0.2652 | 0.6367 | **0.137587** | 13.401025 | 0.2489 | 0.6414 |
| 43 (512, 4, 4) | Normal | 0.122186 | **13.934478** | 0.2793 | 0.6407 | 0.152027 | **12.970874** | 0.2652 | 0.6420 |
| | LS ($\alpha = 0.3$) | **0.139969** | 13.352149 | 0.2765 | 0.6573 | **0.162863** | 12.674551 | 0.2678 | 0.6702 |
| 52 (512, 4, 4) | Normal | 0.162870 | **12.696494** | 0.2735 | 0.6771 | 0.162153 | **12.696784** | 0.2647 | 0.6824 |
| | LS ($\alpha = 0.3$) | **0.168025** | 12.562250 | 0.2756 | 0.7002 | **0.166146** | 12.586002 | 0.2706 | 0.7079 |

Summarizing our findings, we arrive at three practical ways to locate the GPZ:

- **Theory-driven (via $R_c^2$):** use the theoretical criterion based on the intra-class mean-squared radius $R_c^2$ to identify the transition zone.

- **Attack-driven (post hoc from reconstructions):** infer the GPZ location retrospectively from the reconstruction outcomes produced by a model inversion attacker.

- **Training-contrast (one-hot vs. label smoothing):** run MIA against models trained with one-hot labels and with label smoothing, and identify the layer(s) where the LS-trained model's reconstruction MSE begins to rise markedly.

### H.6. Reconstruction of input data based on label smoothing prediction

Moreover, when trained with LS, the model shows a steady increase in MSE regardless of the sign of the smoothing factor $\alpha$, while one-hot training keeps MSE relatively stable. PSNR follows a similar trend. LS with a positive $\alpha$ yields better resistance, consistent with the theory in Section 3. Struppek et al. ([Struppek et al., 2024](#)) also observed that positive smoothing enhances MIA effectiveness in centralized settings, since LS increases prediction entropy, and by Eq. 1, higher

*Table 9.* MIA performance on IR-152's final outputs using different enhancement methods.

| Position | Output Condition | Auxiliary Data Set | | | | Test Data Set | | | |
|---|---|---|---|---|---|---|---|---|---|
| | | MSE | PSNR | SSIM | LPIPS | MSE | PSNR | SSIM | LPIPS |
| Prediction | Normal | 0.170175 | 12.509152 | 0.2760 | 0.7321 | 0.169263 | 12.510290 | 0.2765 | 0.7348 |
| | LS ($\alpha = 0.3$) | 0.165342 | 12.634348 | 0.2763 | 0.6955 | 0.163499 | 12.661634 | 0.2711 | 0.6978 |
| | LS ($\alpha = -0.05$) (Struppek et al., 2024) | 0.171152 | 12.481016 | 0.2777 | 0.7392 | 0.169522 | 12.504991 | 0.2781 | 0.7395 |
| | Log (Yang et al., 2019) | 0.163476 | 12.685201 | 0.2766 | 0.6974 | **0.162863** | **12.676754** | 0.2709 | 0.6953 |
| | Power (1/12) (Zhang et al., 2023) | **0.160062** | **12.757273** | 0.2770 | 0.6951 | 0.163184 | 12.667365 | 0.2704 | 0.6939 |

*Table 10.* Inverse model trained with different auxiliary data attacks the depths of VGG19 trained with MNIST. The table reports reconstruction quality on the auxiliary set and the test set, including MSE, PSNR, SSIM, and LPIPS. **Bold** numbers indicate, for each fixed target block, the maximum MSE / PSNR across auxiliary data sources (MNIST/EMNIST/KMNIST), reported separately for the auxiliary set and the test set.

| Depth (Dim) | Auxiliary Data | Auxiliary Data Set | | | | Test Data Set | | | |
|---|---|---|---|---|---|---|---|---|---|
| | | MSE | PSNR | SSIM | LPIPS | MSE | PSNR | SSIM | LPIPS |
| 17 (256, 16, 16) | MNIST | 0.000383 | 34.171963 | 0.9806 | 0.0217 | 0.000385 | **34.160216** | 0.9807 | 0.0211 |
| | EMNIST | 0.0000850 | **40.748499** | 0.9847 | 0.0042 | **0.000613** | 32.132613 | 0.9539 | 0.0273 |
| | KMNIST | **0.000536** | 32.720340 | 0.9711 | 0.0261 | 0.000395 | 34.036408 | 0.9709 | 0.0326 |
| 26 (256, 16, 16) | MNIST | 0.000962 | 30.178920 | 0.9636 | 0.0366 | 0.001025 | **29.912627** | 0.9624 | 0.0348 |
| | EMNIST | 0.0003603 | **34.461491** | 0.9707 | 0.0126 | **0.001310** | 28.840008 | 0.9242 | 0.0399 |
| | KMNIST | **0.001834** | 27.393657 | 0.9345 | 0.0354 | 0.001194 | 29.241351 | 0.9418 | 0.0417 |
| 39 (512, 8, 8) | MNIST | 0.001687 | **27.740108** | 0.9309 | 0.0361 | 0.004056 | **23.947208** | 0.8980 | 0.0445 |
| | EMNIST | 0.001823 | 27.416396 | 0.9315 | 0.0282 | 0.007511 | 21.265380 | 0.7507 | 0.0770 |
| | KMNIST | **0.007273** | 21.406130 | 0.7778 | 0.0663 | **0.008491** | 20.727759 | 0.7505 | 0.0751 |
| 43 (512, 4, 4) | MNIST | 0.001908 | **27.209829** | 0.9261 | 0.0383 | 0.006320 | **22.031106** | 0.8647 | 0.0539 |
| | EMNIST | 0.004395 | 23.603742 | 0.9010 | 0.0351 | 0.017480 | 17.591984 | 0.6162 | 0.1311 |
| | KMNIST | **0.014448** | 18.429939 | 0.7702 | 0.0766 | **0.021117** | 16.768995 | 0.5984 | 0.1439 |
| 52 (512, 4, 4) | MNIST | 0.028791 | 15.423719 | 0.7583 | 0.0824 | 0.029401 | **15.336118** | 0.6676 | 0.1142 |
| | EMNIST | 0.017865 | **17.512356** | 0.5973 | 0.1403 | 0.056163 | 12.511854 | 0.2169 | 0.3586 |
| | KMNIST | **0.044163** | 13.570527 | 0.6734 | 0.1146 | **0.060255** | 12.210248 | 0.5656 | 0.1639 |

entropy enables stronger MIA. However, as discussed in the threat model, CI inputs are privacy-sensitive. Table 9 compares different entropy-enhancing strategies. Although all three improve reconstruction quality the resulting MSE exceeds 0.16, rendering the recovered inputs unlikely to pose a practical threat.

### H.7. The role of data types

This subsection supplements Part 4 of Section 4.2. Table 10 reports the reconstruction performance on the auxiliary training set and the attack performance on the test set for MNIST, under different auxiliary datasets used to train the inversion model.

A consistent pattern with the KMNIST case is observed. Even when the target dataset undergoes a representational transition, the attack remains relatively effective at Block 43 when the auxiliary data are drawn from the same distribution as the target (i.e., MNIST→MNIST), indicating weaker resistance at this depth under the in-distribution auxiliary setting. In contrast, under approximately matched but out-of-distribution auxiliary data (e.g., EMNIST or KMNIST as auxiliary), Block 43 already exhibits a clear defensive effect: reconstruction quality degrades substantially on the test set compared to the in-distribution case, suggesting that distribution mismatch accelerates the practical onset of inversion failure after the transition.

We further investigate the case of face datasets, which exhibit low intra-class variation. As shown in Fig. 12, reconstructions ultimately converge to a representative frontal face for each class, resembling the behaviour seen in Fig. 9. The consistency of samples within the 530 classes in FaceScrub causes the transition zone to shift earlier; MSE starts rising around Block 26 and climbs steeply after Block 30. Interestingly, although attacking CIFAR-10 used the same distribution, its large intra-class variance leads to poorer final reconstruction compared to both the structured handwriting datasets and the consistent face data. In such cases, reconstructions tend to regress toward the auxiliary data distribution only when the target representation has low variability.

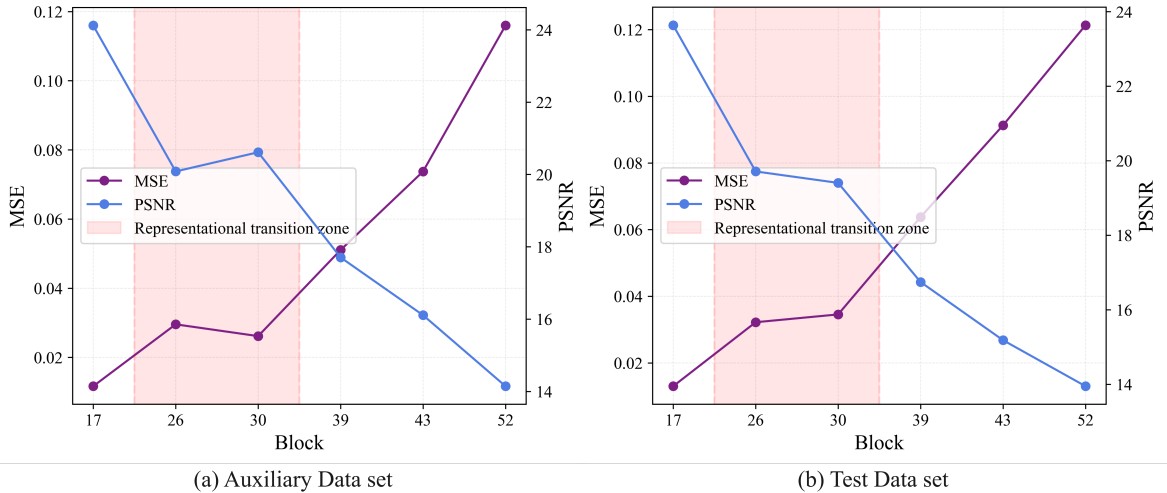

(a) Auxiliary Data set          (b) Test Data set

*Figure 11.* MIA performance across different depths of VGG19 trained with label smoothing on the FaceScrub dataset.

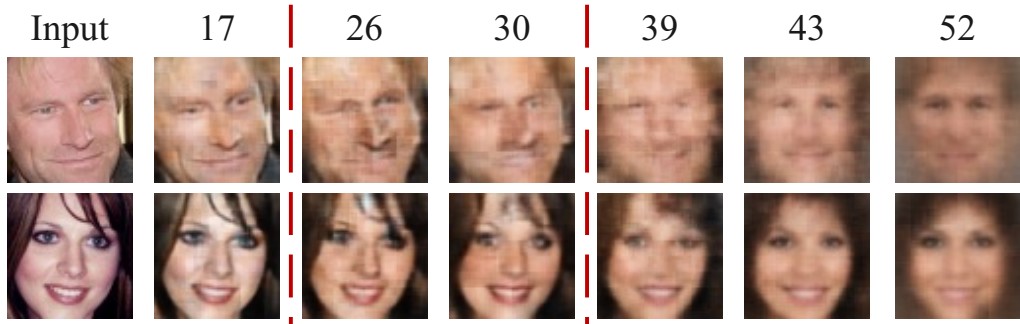

*Figure 12.* MIA visualization performance of different depths of VGG19 face recognizer after LS training. Dotted lines distinguish the representational transition zone.

## I. Deployment overhead

Table 11 decomposes a CI split into three practical cost families: (i) **Latency** on the edge, (ii) **Communication & Memory Overhead**, and (iii) **Energy**. The goal is not to optimize any single metric in isolation, but to expose the dominant bottlenecks that constrain real CI deployments and to quantify the tradeoffs across split points.

All CPU-side measurements use a fixed cap of 8 threads. Latency statistics are obtained with a warmup phase of $N_{\text{warm}}^{\text{lat}} = 50$ iterations, which are excluded, followed by $N_{\text{lat}} = 200$ timed forward passes; mean and p95 are computed over these per-iteration wall-clock latencies. Communication and memory metrics are derived deterministically from the activation tensor shapes and model footprints under the same input resolution and split definition, and therefore do not depend on the iteration count. Energy measurements use a longer window to reduce variance due to RAPL readout granularity/noise: we run $N_{\text{warm}}^{\text{eng}} = 100$ warmup iterations, which are excluded, then measure over $N_{\text{eng}} = 500$ iterations with the same 8-thread setting; we report $E_{\text{inf}} = E_{\text{total}}/N_{\text{eng}}$ with $N_{\text{eng}} = 500$.

### I.1. Latency

**What the metrics mean.** Edge FLOPs and CPU latency (mean/p95) characterize the edge-side execution cost of the trunk up to the split point. **FLOPs** (floating-point operations) denote the total number of arithmetic operations required by the trunk for one forward pass under the given input resolution and split configuration. In our accounting, convolution and linear layers dominate; non-linearities and simple elementwise ops (e.g., ReLU, pooling, normalization) contribute comparatively little. We follow the common convention that one multiply–accumulate (MAC) counts as 2 FLOPs (one multiply + one add), so FLOPs scale roughly with the amount of dense compute. FLOPs are hardware-agnostic and useful for comparing

*Table 11.* Edge-side deployment costs under CI splits (batch=1). **Bold** indicates the largest value within each row. Tx reports the raw tensor size assuming FP32; FP16/INT8 scales linearly. Params/Buffers are static FP32 memory. Act peak is the peak live activation tensor footprint (tensor-only). Energy metrics are measured via CPU RAPL on the same host; EDP/ED$^2$P are computed over the same measurement protocol with a fixed number of iterations (warmup excluded).

| | VGG (depth-26) | VGG (depth-30) | IR152 (block-49) |
|---|---|---|---|
| **Latency** | | | |
| Output shape | **(1,256,16,16)** | (1,512,8,8) | (1,512,4,4) |
| Edge FLOPs (GFLOPs) | 1.826 | 1.977 | **7.727** |
| CPU latency (ms, mean/p95) | 2.60 / 4.39 | 2.65 / 4.81 | **16.11 / 24.84** |
| **Communication & Memory Overhead** | | | |
| Tx (FP32, KB) | **256** | 128 | 32 |
| Params (MiB, FP32) | 8.88 | 13.39 | **184.81** |
| Total params (count) | **139,622,218** | 139,622,218 | 62,091,978 |
| Edge params (count/share) | 2,328,384 (1.67%) | 3,509,568 (2.51%) | **48,373,696 (77.91%)** |
| Buffers (MiB) | 0.011 | 0.015 | **0.172** |
| Act peak (MiB, tensor-only) | 1.00 | 1.00 | **1.25** |
| **Energy** | | | |
| Energy (J) total | 498.6036 | 623.9095 | **3143.8618** |
| Energy per inf (J/inf) | 0.9972 | 1.2478 | **6.2877** |
| GFLOPs/W | **0.003662** | 0.003169 | 0.002458 |
| EDP | 628.1730 | 1076.8269 | **26785.0955** |
| ED$^2$P | 791.4130 | 1858.5325 | **228203.8402** |

compute across split points, but they do not capture platform effects such as memory bandwidth, cache locality, vectorization, kernel launch overhead, and thread scheduling. Therefore, **CPU latency** is the deciding system metric. Mean latency reflects typical responsiveness, while p95 captures tail latency and jitter, which are critical for low-latency interactive CI. The output shape $Z_\Delta$ indicates the activation tensor that will be produced at the split and then stored/transmitted.

**How to read high/low values.** Lower mean/p95 indicates better real-time feasibility. Lower FLOPs usually correlates with lower latency on the same platform, but the correlation is imperfect; latency reflects both compute and memory-system behavior, so it should be treated as the primary feasibility indicator.

**What the table says quantitatively.** Compared with VGG depth-26, the IR152 split increases edge compute by $7.727/1.826 \approx 4.23\times$ and mean latency by $16.11/2.60 \approx 6.20\times$ (p95: $24.84/4.39 \approx 5.66\times$). Within VGG, moving from depth-26 to depth-30 changes mean latency only by $2.65/2.60 \approx 1.017\times$ ($\sim$1.7%), indicating that depth-30 is nearly latency-neutral relative to depth-26 under this measurement.

### I.2. Communication & Memory Overhead

This block answers two deployment questions: (i) how expensive it is to transmit $Z_\Delta$ and (ii) whether the edge device can hold and run the trunk reliably within memory constraints.

**What the metrics mean.**

- **Tx (FP32)** is the raw payload size of $Z_\Delta$ if serialized in FP32. It is computed from the tensor shape as $|Z_\Delta| \times 4$ bytes, and therefore scales linearly with precision: FP16 halves Tx and INT8 quarters it. Tx directly affects uplink time, networking energy, and end-to-end CI latency when bandwidth is limited.

- **Params/Buffers** measure the static memory footprint of the edge trunk. This is the dominant contributor to model storage and can constrain deployment on memory-limited devices.

- **Act peak** is the peak live activation memory during trunk execution (tensor-only). It approximates the minimum runtime memory budget required to avoid out-of-memory failures during inference.

**How to read high/low values.** Lower Tx is beneficial when network bandwidth or uplink latency is the bottleneck. Lower

Params/Buffers is beneficial when the edge device has limited RAM/flash or when model loading and caching overhead matters. Act peak matters when runtime memory headroom is tight.

**What the table says quantitatively.** Within VGG, moving from depth-26 to depth-30 halves the transmitted payload (256KB→128KB, a $2\times$ reduction), while increasing the edge-side parameter footprint by $13.39/8.88 \approx 1.51\times$. IR152 achieves a smaller Tx (32KB), which is $256/32 = 8\times$ smaller than VGG depth-26 and $128/32 = 4\times$ smaller than VGG depth-30, but this comes with a substantially larger edge model: Params are $184.81/8.88 \approx 20.8\times$ (vs. VGG depth-26) and $184.81/13.39 \approx 13.8\times$ (vs. depth-30). Notably, Act peak differs only mildly (1.00–1.25 MiB), so the dominant footprint gap is the edge trunk model size rather than activations.

**Deployment implication.** If the edge is bandwidth-limited, deeper splits that reduce Tx can be attractive, but only if the edge device can afford the larger trunk (Params/Buffers). Table 11 shows that VGG depth-30 provides a strong "Tx reduction with minimal latency change" option, whereas the IR152 deep split reduces Tx further but makes the trunk footprint much heavier.

### I.3. Energy

Energy metrics determine whether a split is viable in always-on, thermally constrained, or battery-limited settings. They also reflect latency from a different angle because energy accumulates with sustained execution time.

**What the metrics mean.**

- **Energy total (J)** is the total CPU energy consumed over the measurement window (fixed protocol).

- **Energy per inf (J/inf)** is the amortized energy per inference, typically the most deployment-relevant quantity for battery-limited or always-on CI.

- **GFLOPs/W** is achieved compute throughput per watt under the same FLOPs accounting. Since FLOPs are a model-side proxy, GFLOPs/W is most meaningful for relative comparison across split points on the same platform and measurement protocol.

- **EDP** (J·s) and **ED$^2$P** (J·s$^2$) jointly penalize energy and delay. ED$^2$P penalizes slow execution more strongly and is useful when tail latency is unacceptable.

**How we compute energy metrics.** We run warmup (excluded), then measure over $N$ inference iterations. Let $E_d^{(0)}$ and $E_d^{(1)}$ be RAPL energy readings (Joules) for domain $d$ at start/end, and $\Delta E_d = E_d^{(1)} - E_d^{(0)}$. Then:

$$E_{\text{total}} = \sum_{d \in \mathcal{D}} \Delta E_d, \qquad T = t_1 - t_0, \qquad P_{\text{avg}} = \frac{E_{\text{total}}}{T}, \qquad E_{\text{inf}} = \frac{E_{\text{total}}}{N}, \tag{55}$$

$$\text{GFLOPs/W} = \frac{(\text{FLOPs}/T)/10^9}{P_{\text{avg}}}, \qquad \text{EDP} = E_{\text{total}} \cdot T, \qquad \text{ED}^2\text{P} = E_{\text{total}} \cdot T^2. \tag{56}$$

**How to read high/low values.** Lower $E_{\text{inf}}$ means lower energy cost per edge inference. Lower EDP/ED$^2$P indicates better joint energy–latency efficiency; even if average power is similar, a method can consume substantially more energy if it runs longer (since $E_{\text{inf}} \approx P_{\text{avg}} \times$ latency).

**What the table says quantitatively.** IR152 consumes $6.2877/0.9972 \approx 6.31\times$ more energy per inference than VGG depth-26 and $6.2877/1.2478 \approx 5.04\times$ more than VGG depth-30. The gap is magnified in energy–delay metrics: EDP is $26785/628 \approx 42.6\times$ (vs. depth-26) and $26785/1077 \approx 24.9\times$ (vs. depth-30), and ED$^2$P is $228204/791 \approx 288\times$ (vs. depth-26) and $228204/1859 \approx 123\times$ (vs. depth-30), highlighting that deep residual splits can be particularly unfavorable when low-latency operation is required. Within VGG, moving from depth-26 to depth-30 increases $E_{\text{inf}}$ by $1.2478/0.9972 \approx 1.25\times$ while halving Tx, offering a practical "spend ~25% more energy to halve communication" tradeoff knob.

## J. Related Work

Model inversion attacks in collaborative inference were first explored by He et al. (He et al., 2019), who demonstrated that shallow feature representations could be exploited to reconstruct input data. Subsequent research observed that increasing

model depth helps resist such attacks. Liu et al. (Liu et al., 2025) further clarified that the success of MIA is governed by information-theoretic factors such as mutual information, conditional entropy and effective information content.

During this period, advancements in attack methodology were relatively slow, as the original decoder architectures were already effective in reconstructing inputs from shallow features (Yang et al., 2019). Notable developments include Yin et al. (Yin et al., 2023), who proposed a generative approach capable of performing attacks without auxiliary data, and Chen et al. (Chen et al., 2024), who introduced a conditional diffusion-based method to approximate the inverse of forward models.

In contrast, defence strategies have seen more rapid innovation. Representative works include Wang et al. (Wang et al., 2022), who proposed deepening the model dynamically and adding differential privacy noise; Xia et al. (Xia et al., 2025) and Duan et al. (Duan et al., 2023), who introduced information-theoretic defences by maximizing conditional entropy and estimating mutual information via CLUB (Contrastive Log Upper Bound), respectively; and Azizian et al. (Azizian & Bajić, 2024), who incorporated an autoencoder architecture to increase inversion resistance.

Across these defences, two common patterns emerge: first, many approaches increase model depth, which can facilitate the transition from feature- to decision-level representations; second, they aim to increase $H(X \mid Z)$ or reduce $H(Z)$, in line with the theoretical framework in (Liu et al., 2025). However, these defences often incur accuracy degradation and do not explicitly target the essence of representational transition. Moreover, most existing works still place the partition point in shallow layers, overlooking the central role of split location in CI.

Another important observation highlighted in prior CI defence discussions is that protocol-level cryptographic protections (e.g., HE and secure computation) can in principle hide transmitted representations from an untrusted cloud, but they often introduce substantial computation and communication overhead, which can be challenging, and often impractical, under the real-time and resource-constrained assumptions typical in CI. For example, early CI work explicitly notes that homomorphic encryption suffers from "huge inefficiency" and is not applicable to all DNN operations (He et al., 2019). Moreover, while the broader private inference literature has made progress on optimizing encrypted inference (e.g., 2PC-based cryptographic inference services and HE-based encrypted split inference) (Srinivasan et al., 2019; Pereteanu et al., 2022; Elkhatib et al., 2026), CI–encryption integration remains comparatively under-explored and is typically evaluated under limited model scales or constrained deployment settings, with end-to-end latency and system complexity still being major barriers for low-latency edge–cloud CI. Rather than proposing yet another defence, we address the upstream split-location problem and identify a theoretically grounded partition zone that provides intrinsic resistance; this guidance can inform future work on designing stronger defences, developing more targeted attacks, and making practical deployment choices in CI.

**Scope clarification.** Our goal is not to propose an incremental defence that perturbs representations, but to characterize and predict intrinsic inversion resistance in CI, and to provide actionable guidance on split selection that is complementary to existing defences. Therefore, the appropriate evaluation is theory validity, cross-architecture verification, and robustness under strengthened attackers, rather than a head-to-head comparison with a particular defence.

---

**Algorithm 1** Locating the Golden Partition Zone (dimension-normalized)

---

**Require:** Input batch $X = \{x_i\}_{i=1}^B$, labels $\{y_i\}$, edge trunk $f_{\text{edge}}$, target layer list $L_{\text{target}} = [l_1, \ldots, l_T]$ (shallow→deep), threshold $\tau$ (e.g., 20%)

**Ensure:** Transition zone: from $l_{\text{TS}}$ to $l_{\text{TP}}$

1: **function** EXTRACT($f_{\text{edge}}, X, l$)
2:     **return** representations $Z^{(l)} = \{z_i^{(l)}\}_{i=1}^B$ at layer $l$         ▷ e.g., via forward hooks
3: **end function**
4: **for** $t = 1$ to $T$ **do**
5:     $l \leftarrow l_t$
6:     $Z^{(l)} \leftarrow$ EXTRACT($f_{\text{edge}}, X, l$)
7:     $d_l \leftarrow \dim(z^{(l)})$         ▷ flattened representation dimension
8:     **if** All samples belong to the same class $c$ **then**
9:         $\mu_c^{(l)} \leftarrow \frac{1}{B} \sum_{i=1}^B z_i^{(l)}$
10:         $R_c^{2(l)} \leftarrow \frac{1}{B} \sum_{i=1}^B \|z_i^{(l)} - \mu_c^{(l)}\|^2$
11:     **else**
12:         Initialize $S \leftarrow 0$, $C \leftarrow$ number of classes present in the batch
13:         **for** each class $c$ in the batch **do**
14:             $\mathcal{I}_c \leftarrow \{i : y_i = c\}$, $N_c \leftarrow |\mathcal{I}_c|$
15:             $\mu_c^{(l)} \leftarrow \frac{1}{N_c} \sum_{i \in \mathcal{I}_c} z_i^{(l)}$
16:             $R_c^{2(l)} \leftarrow \frac{1}{N_c} \sum_{i \in \mathcal{I}_c} \|z_i^{(l)} - \mu_c^{(l)}\|^2$
17:             $S \leftarrow S + R_c^{2(l)}$
18:         **end for**
19:         $R^{2(l)} \leftarrow \frac{1}{C} S$         ▷ class-averaged intra-class radius
20:     **end if**
21:     $\widetilde{R}^{2(l)} \leftarrow \frac{R^{2(l)}}{d_l}$         ▷ dimension normalization
22: **end for**
23: **for** $t = 2$ to $T$ **do**
24:     $l_{\text{prev}} \leftarrow l_{t-1}$, $l_{\text{cur}} \leftarrow l_t$
25:     $R_{\text{drop}}^{2(l_t)} \leftarrow \frac{\widetilde{R}^{2(l_{\text{prev}})} - \widetilde{R}^{2(l_{\text{cur}})}}{\widetilde{R}^{2(l_{\text{prev}})}} \times 100\%$
26: **end for**
27: $l_{\text{TP}} \leftarrow \arg\max_{t \in \{2,\ldots,T\}} R_{\text{drop}}^{2(l_t)}$
28: $l_{\text{TS}} \leftarrow \arg\max_{t < \text{index}(l_{\text{TP}})} R_{\text{drop}}^{2(l_t)}$         ▷ largest drop before $l_{\text{TP}}$
29: $L_{\text{intermediate}} \leftarrow$ layers between $l_{\text{TS}}$ and $l_{\text{TP}}$ in $L_{\text{target}}$
30: **if** $\forall l \in L_{\text{intermediate}}$, $|R_{\text{drop}}^{2(l)}| < \tau$ **then**
31:     **return** transition at $l_{\text{TP}}$
32: **else**
33:     **return** transition zone from $l_{\text{TS}}$ to $l_{\text{TP}}$
34: **end if**

---

