# OpenReview forum: "Partitioning for Intrinsic Model Inversion Resistance in Collaborative Inference"
_ICML.cc/2026/Conference — ICML 2026 regular_

### Official Review · Reviewer_MWZy · 2026-03-13

**Soundness:** 3
**Presentation:** 3
**Significance:** 3
**Originality:** 3
**Overall Recommendation:** 5
**Confidence:** 2

**Summary:**

This paper studies a key problem in collaborative inference: at which layer should a model be partitioned so that it is naturally more resistant to model inversion attacks. The authors challenge the intuition that deeper always means safer, and argue that the real determining factor is not depth itself, but whether the intermediate representation has transitioned from feature-level to decision-level. Based on this perspective, the paper introduces the Golden Partition Zone, or GPZ, together with the corresponding layer-wise metric $R_c^2$, to identify more privacy-friendly partition layers, and validates this idea across multiple vision architectures.

**Compliance With Llm Reviewing Policy:**

Affirmed.

**Final Justification:**

My main concern in the initial review was the positioning of Neural Vortex; given the evidence provided, it felt somewhat over-mechanized. In the rebuttal, the authors clarified that this concept is intended as an auxiliary interpretive perspective rather than a core theoretical contribution. This addressed my main concern and made my overall evaluation more positive.

**Key Questions For Authors:**

I do not see any major issues with the paper.

**Limitations:**

Yes

**Strengths And Weaknesses:**

Strengths

1. The paper targets a core privacy issue in collaborative inference.  The paper focuses on a more fundamental and deployment-relevant question: at which layer should a model be partitioned so that it is naturally more resistant to model inversion attacks, rather than relying only on additional defense modules or noise mechanisms.

2. The central claim is novel and challenges the common intuition that depth implies security. The most distinctive point of the paper is that depth itself is not the root cause of improved inversion resistance. What really matters is whether the intermediate representation undergoes a transition from feature-level to decision-level. Through comparisons across architectures such as ViT, ResNet, and VGG, the paper shows that deeper is not always safer, and that the key issue is whether the representation in a certain region shifts from retaining substantial instance-level information to becoming more decision-oriented.

3. The theoretical narrative is relatively complete and is well connected to the empirical observations.  The paper does not merely report an empirical phenomenon, but also attempts to explain it from an information-theoretic perspective. For example, it discusses how mutual information decays with depth while the decay rate depends on architecture, and how the lower bound of conditional entropy may change significantly in the transition region. This makes the paper more than an observation of a pattern and gives it a stronger explanatory component.

4. The experiments cover multiple backbones with substantially different structures.  The paper does not evaluate only a single architecture, but includes IR-152, ResNet-50, VGG19, and ViT. Since these models differ substantially in their representation mechanisms, this helps support the central claim that architecture influences both the transition and the appropriate partition boundary.

Weaknesses

1. The notion of “Neural Vortex” feels somewhat packaged, and the underlying mechanism still needs stronger evidence.  At present, this concept appears more like a summary of a set of observed training dynamics than a fully established mechanism with clear necessity and generality. Rather than a rigorously grounded new theoretical object, it reads more like a conceptual label for an empirical phenomenon. If the authors intend this notion to be a major contribution of the paper, they may need to provide more direct evidence showing when it arises, whether it is stable, and whether it has a clear causal connection to the improvement in inversion resistance.

---

> ### Author Rebuttal · Authors · 2026-03-26
>
> We thank the reviewer for the positive assessment of our work, the time devoted to the review, and the constructive comments on the Neural Vortex (NV) concept.
>
> We would first like to clarify that, in the original intent of the manuscript, NV is introduced as a supporting explanatory concept for training dynamics rather than a standalone formal mechanism. It is not intended as a purely rhetorical label, but as a compact explanatory concept grounded in the theoretical derivations and analysis developed in the paper. We also recognize that the current wording may invite over-interpretation of NV as a fully formalized theoretical object. In the revision, we will make its intended role explicit and revise the presentation accordingly. Our specific responses are as follows.
>
> **Contribution positioning.** The core contribution of this paper lies in the proposed Golden Partition Zone (GPZ) and the $R_c^2$-based characterization of representation transition, which together guide partition decisions in collaborative inference. By contrast, NV is introduced to interpret how training dynamics, particularly label-distribution design, affect the evolution of $R_c^2$, and thereby influence inversion resistance. In the revision, we will make this distinction more explicit and clearly position NV as a supporting explanatory concept rather than a primary theoretical object.
>
> **Naming.** The term “Neural Vortex” is intentionally metaphorical, as noted in Appendix G. It is introduced to summarize the dynamic intra-class contraction behavior characterized in our analysis at a conceptual level, rather than to define a new formal theoretical construct. We will further refine the wording to ensure that this intended role is conveyed more clearly and cannot be misread as a claim of an independent formal mechanism.
>
> **Analytical grounding and interpretive chain.** Importantly, NV is not introduced as a purely descriptive label; rather, it is a compact explanatory concept grounded in the analytical framework developed in the paper. We provide a first-order characterization of the evolution of $R_c^2$ (Eqs. (9)–(11) and Appendix E), and connect it to the lower bound on $H(X|Z)$ via Eq. (8), thereby linking representation contraction, uncertainty about the input given $Z$, and inversion difficulty. Empirically, across multiple architectures, models trained with label smoothing consistently exhibit stronger inversion resistance in the decision-level representation zone.
>
> Regarding the reviewer’s concern about conditions, stability, and causal connection, the paper provides analysis-backed and empirically consistent evidence for the following interpretation: label-distribution design affects the evolution of $R_c^2$. Under the locally consistent T-term sign conditions analyzed in Sec. 3.2 and Appendix E.4, $R_c^2$ tends to contract in the decision-level representation zone and, through the lower bound on $H(X|Z)$, this contraction is associated with stronger inversion resistance. This behavior is consistently observed across the considered architectures under a unified training setup. Accordingly, we frame NV as a scope-limited, analysis-backed explanatory perspective on the training dynamics analyzed in this study.
>
> **Revision.** We appreciate the reviewer’s suggestion. In the revision, we will revise the relevant text to (i) explicitly distinguish GPZ from NV in the contribution statement, (ii) describe NV consistently as an explanatory concept in the main text and appendix, and (iii) avoid wording that may suggest that NV is an independently formalized mechanism or theory.

---

> > ### Author Rebuttal · Reviewer_MWZy · 2026-04-01
> >
> > Thank you for the rebuttal. The authors have addressed all of my questions and provided sufficient clarification on my main concerns. Based on the response, I am revising my score to 5.

---

### Official Review · Reviewer_AW62 · 2026-03-14

**Soundness:** 3
**Presentation:** 3
**Significance:** 3
**Originality:** 3
**Overall Recommendation:** 4
**Confidence:** 4

**Summary:**

The paper describes strategies to decide where to partition in collaborative inference settings, so as to minimize the success of model inversion attacks. Their analysis is grounded in information theory, and the experimental results support the findings.

**Compliance With Llm Reviewing Policy:**

Affirmed.

**Final Justification:**

The response and discussion with both myself and other reviews re-iterates the value of this paper; I maintain my support for it.

**Key Questions For Authors:**

Please look at weaknesses and respond to them!

**Limitations:**

Yes

**Strengths And Weaknesses:**

Strengths:

1. Analysis is quite interesting. While intuitive, this paper's contributions are valuable because of the theoretical grounding.

2. The paper tackles a problem that has received limited attention in the recent past. While the authors show cherry picked examples of where CI is used, it is still not mainstream for reasons I shall discuss next.

3. The paper is well written; experiments are thoughtful and communicate the point well.

Weaknesses:

1. There are little details about the attacks used for reconstruction. Is it the case that these off-the-shelf attacks are not optimally tuned for this task? In theory, while the amount of information shared is more sparse, sufficient optimization should be able to reconstruct the image (under the assumption that the reconstruction module is provided some prior information).

2. There is little discussion on the trade-off b/w the chosen depth and the amount of information that has to be shared (i.e., dimensionality at that layer). The latter is also a dominant factor essential in determining where to split.

3. The chosen perceptibility metrics also don't particularly highlight semantic similarity or human perception; one has to include more metrics to highlight whether the results are robust in this regime.

4. Little discussion placed on how this phenomenon extends to the transformer architecture in the text domain.

---

> ### Author Rebuttal · Authors · 2026-03-27
>
> We thank the reviewer for the time and effort devoted to evaluating our paper and for the constructive comments. We respond to the specific concerns below.
>
> # Attack
> We agree that the attack details were not sufficiently emphasized in the main text, although they are described in Secs. 4.2(2)–4.2(3) and Appendices H.3–H.4, including the inversion models, training procedures, and attack enhancement strategies.
>
> We also agree that a single off-the-shelf inversion attack should not be treated as the leakage upper bound. To address this concern, we do not rely on a fixed off-the-shelf decoder alone, but strengthen the attacks from two directions as stress tests.
>
> 1. On the representation side, we introduced H(z) enhancement strategies (FFT-based, normalization-based, and NN-based augmentations) to mitigate information loss and optimization difficulty when the shared representations are low-information and relatively sparse.
>
> 2. We designed stronger inversion models such as Inversion-IR152, which can be viewed as a mirror-style enhanced attack when the attacker has prior knowledge of the edge-side architecture. We also combined multiple attention mechanisms to improve inverse fitting and recovery.
>
> More importantly, Tables 3 and 4 show that, even under these stronger and combined settings, decision-level representations remain substantially more resistant to inversion. This indicates that our conclusion does not rely on any single fixed and potentially weak off-the-shelf decoder, but continues to hold under stronger attack stress tests.
>
> Theoretically, our conclusion is supported not only by the empirical performance of a particular attack model, but also by the intrinsic difficulty characterization from our information-theoretic analysis. The recoverability of the input from the shared representation is constrained by H(X|Z). Stronger attacks may improve reconstruction, but they do not remove the information-theoretic disadvantage of decision-level representations under the same threat model.
>
> # Trade-off
> We agree that this point was not sufficiently emphasized in the main text, although related results and discussion are already included in the original manuscript. Specifically, Tables 2 and 6 report the intermediate output dimensions at different partition depths, while Sec. 5 and Appendix I discuss split trade-offs from a deployment perspective, including dimensionality as well as practical factors such as latency, communication and memory overheads, and energy.
>
> In addition, because the reviewer specifically emphasizes the role of dimensionality, Appendix H.5 in the original manuscript already provides a targeted analysis using ResNet-50. Since the bottleneck blocks in ResNet-50 involve a typical 4× output-dimension expansion, we observed that when depth increases while dimensionality remains unchanged, MIA becomes slightly weaker; when the split crosses a bottleneck, reconstruction quality improves again (see Fig. 10). This shows that dimensionality does affect inversion difficulty, consistent with the roles of d and D in Eq. (8). At the same time, once the representation enters the decision-level zone, the gain from increased dimensionality becomes much weaker. This suggests that dimensionality matters, but does not by itself determine overall inversion resistance or split quality.
>
> We will make this trade-off discussion clearer in the revised main text and explicitly highlight Appendix H.5 as a targeted supplementary analysis of dimensionality.
>
> # Metric
> We thank the reviewer for this valuable suggestion and agree that adding more metrics is important. Therefore, we have added SSIM and LPIPS to the revised Tables 7–10 (https://anonymous.4open.science/r/Supplement-200D). Specifically, LPIPS is widely used for perceptual similarity and has been shown to align better with human judgments than traditional metrics [1]. Importantly, the results under these added metrics are consistent with our original conclusions.
>
> > [1] Zhang R, Isola P, Efros A A, et al. "The unreasonable effectiveness of deep features as a perceptual metric." *Proceedings of the IEEE conference on computer vision and pattern recognition*. 2018: 586-595.
>
> # Scope extension
> We appreciate the reviewer’s point. Because the current scope of the paper is explicitly focused on images and deep vision models, we are cautious about making strong claims regarding the text domain without direct empirical evidence.
>
> That said, the manuscript does include a dedicated analysis of Vision Transformer. Since a fixed number of patch tokens continuously preserve feature information across depth, a clear representational transition is unlikely to emerge. This observation motivates a cautious hypothesis for text Transformers: if deeper hidden states also preserve token-level lexical or local information, then a clear GPZ may likewise be difficult to form.
>
> We will clarify this discussion in the revised manuscript and present it as a direction for future work.

---

> > ### Author Rebuttal · Reviewer_AW62 · 2026-04-02
> >
> > Thanks for your response. I remain enthusiastic about the paper, but will wait for discussion with the other reviewers before updating the score.

---

### Official Review · Reviewer_bELR · 2026-03-14

**Soundness:** 3
**Presentation:** 3
**Significance:** 3
**Originality:** 3
**Overall Recommendation:** 4
**Confidence:** 4

**Summary:**

The paper studies the problem of how fast the information about the input propagate through the layers of the network. Solution to this question allows understanding whether reduced information leakage about the input is possible while not exceeding the compute constraint of the edge user in collaborative inference. The authors prove a lower bound for the mutual information between input and intermediate representations, that relates to intra-class mean-squared radius. The authors further empirically showed interesting architectural and algorithmic insights such as:
1. Label smoothing in training reduces intra-class mean-squared radius and results in smaller mutual information lower bound, thus reducing information leakage.
2. Certain architectures (VGG-like backbones) incur Neural Vortex where the leakage suddenly reduces at one shadow layer, thus being more privacy friendly. By contrast, certain other architectures (ViT-style models) induce smoother leakage change across layers and are not ideal for split inference with privacy protection.

**Compliance With Llm Reviewing Policy:**

Affirmed.

**Final Justification:**

Thanks for the discussions. I have no further questions. The data dependency question seems subtle, and may challenge real-world usage a little bit since data distribution drifts are common. Nevertheless, the paper as-is provides valuable insights for better partition layer from the data and architectural perspectives. I'll maintain the positive rating.

**Key Questions For Authors:**

Main questions are on how the conclusion hold for potential worst-case (user) data distribution, see weakness for details.

**Limitations:**

Yes

**Strengths And Weaknesses:**

Strengths: Clear motivation and information theoretic relation between information leakage and intra-class radius, which is also supported by extensive experiments of inversion attack results under various model architectures.

My main remaining concerns (questions) are:
1. The leakage lower bound is with regard to a data distribution of X, how sensitive is the conclusion to different data distributions? Specifically, how does the leakage lower bound or evolution of intra-class radius look like for the worst-case user data (distribution)? Do we need larger number of layers before splitting for worst-case user?
2. The experiments are for one inversion attack and may only reflect artifacts of suboptimal reconstruction attacks rather than ground-truth leakage. It would be nice if the authors could offer additional discussions on how to estimate the lower bound, i.e., the intra-class radius.

---

> ### Author Rebuttal · Authors · 2026-03-26
>
> We thank the reviewer for the time and effort devoted to evaluating our paper and for the constructive comments. We respond to the specific concerns below.
>
> # Data dependence
> This is a very insightful question, as it goes to the core of how data type affects where the representation enters the decision-level zone. Building on our current results, we can further discuss this point through three comparative cases, and we will add this discussion explicitly in the revision.
>
> (i) Compared with MNIST, KMNIST, and FaceScrub, CIFAR-10 exhibits larger intra-class diversity in our observed settings, while having only 10 classes. In this case, the transition is more gradual and the zone is wider (Fig. 6).
>
> (ii) By contrast, FaceScrub has relatively smaller intra-class variation but many classes. In our results, the transition appears earlier and is more concentrated, yielding a narrower zone (Fig. 12). Appendix H.7 also shows that the transition shifts earlier for FaceScrub, with MSE starting to rise around Block 26 and increasing steeply after Block 30.
>
> (iii) MNIST and KMNIST show another pattern: they have relatively low visual variance, fewer classes than FaceScrub, and many zero-valued pixels. As discussed in Sec. 4.2, this causes feature extraction to persist across more layers, shifting the GPZ deeper and making the transition zone the narrowest among the settings we studied. Relatively speaking, this is the closest case to a worst-case distribution among the datasets we studied, and therefore requires placing more layers on the edge side before intrinsic resistance emerges.
>
> Theoretically, a user distribution that is even more adverse than case (iii) would further worsen the situation along two dimensions in Eq. (8). First, a smaller $H(X)$ directly lowers the bound on $H(X \mid Z)$, thus weakening the theoretical guarantee of intrinsic resistance. Second, if the user distribution causes same-class representations to remain less concentrated at the candidate split layer, then the corresponding $R_c^2$ becomes larger, which also lowers the bound. Under this more adverse condition, inversion may become easier, the onset of intrinsic resistance is expected to occur later, and a deeper split may be required before it appears. This is qualitatively consistent with the visual pattern observed for KMNIST in Fig. 9.
>
> In the current manuscript, our main emphasis was on showing that the transition boundary is data-dependent and that data types affect inversion behaviour. Your comment is very valuable because it points us to make this implication more explicit in Sec. 5. We will therefore expand the discussion there, which we believe will further improve the completeness of the paper.
>
> # Inversion attack
> We agree that no single inversion architecture should be directly equated with the ground-truth leakage itself. However, our experiments are not limited to one fixed or weak reconstruction attack. Instead, in the current manuscript, we already stress-test inversion resistance through two attack-strengthening directions.
>
> 1. In Sec. 4.2(2), we strengthen the attack setting on the representation side, with the goal of enriching the intermediate information H(z), through FFT-based, normalization-based, and NN-based augmentation strategies; details are provided in Appendix H.4.
>
> 2. In Sec. 4.2(3), we strengthen the inversion model itself through progressively enhanced architectures, including MHA, Attention-as-Conv, SE, LSK, MSCA, and a reverse-structured residual variant.
>
> Across these settings, a consistent pattern is shown in Tables 3 and 4: these enhancements are substantially more effective in the feature-representation zone, whereas the decision-level zone remains much more resistant to inversion. Therefore, our experiments should be understood not as relying on a single suboptimal decoder, but as strengthened stress tests consistent with the theory-guided distinction between feature-level and decision-level representations.
>
> More importantly, our claim is not built on attack performance alone; the theoretical analysis provides the primary basis, while the strengthened inversion experiments serve as analysis-guided empirical stress tests.
>
> # Estimation
> Thanks for the comment. Practically, the full lower bound in Eq. (8) is difficult to quantify directly. However, as the reviewer pointed out, its key quantity is $R_c^2$, which is exactly what we compute in practice. Its explicit formula is already given below Eq. (6):
> $$
> R_c^2=\frac{1}{N_c}\sum_{i:y_i=c}\lVert z_i-\mu_c\rVert^2,\qquad
> \mu_c=\frac{1}{N_c}\sum_{i:y_i=c} z_i.
> $$
> For each layer, this only requires computing the class mean and the average squared distance of same-class samples to that center, with numerical examples provided in Fig. 3.
>
> Appendix A and E provide the corresponding detailed analysis and discussion, and Appendix D further builds on this with an active localization algorithm based on layer-wise $R_c^2$. We will clarify this in the revision.

---

> > ### Author Rebuttal · Reviewer_bELR · 2026-04-03
> >
> > Thanks for the discussions. I have no further questions. The data dependency question seems subtle, and may challenge real-world usage a little bit since data distribution drifts are common. Nevertheless, the paper as-is provides valuable insights for better partition layer from the data and architectural perspectives. I'll maintain the positive rating.

---

### Decision · Program_Chairs · 2026-04-30

**Decision:**

Accept (regular)

**Comment:**

The paper considers the question of how to partition the model in collaborative inference in a way that resists membership inference attacks. All reviewers participated in the rebuttal and recommended acceptance.
The reviewers agree that the problem is well-motivated, and the paper provides valuable theoretical insights supported by thoughtful experiments. Hence, I recommend acceptance.